CERN-TH-2022-016

# Gravitational Regge bounds

Kelian Häring[a,b] and Alexander Zhiboedov[a]

[a] CERN, Theoretical Physics Department, CH-1211 Geneva 23, Switzerland and
[b] EPFL, Route de la Sorge, CH-1015 Lausanne, Switzerland

We review the basic assumptions and spell out the detailed arguments that lead to the bound on the Regge growth of gravitational scattering amplitudes. The minimal extra ingredient compared to the gapped case - in addition to unitarity, analyticity, subexponentiality, and crossing - is the assumption that scattering at large impact parameters is controlled by known semi-classical physics. We bound the Regge growth of amplitudes both with the fixed transferred momentum and smeared over it. Our basic conclusion is that gravitational scattering amplitudes admit dispersion relations with two subtractions. For a sub-class of smeared amplitudes, black hole formation reduces the number of subtractions to one. Finally, using dispersion relations with two subtractions we derive bounds on the local growth of relativistic scattering amplitudes. Schematically, the local bound states that the amplitude cannot grow faster than $s^2$.

## CONTENTS

## I. INTRODUCTION AND SUMMARY OF RESULTS

Nonperturbative, relativistic scattering amplitudes in gapped theories admit dispersion relations with at most two subtractions. This is based on the following bound on the Regge growth of the $2 \to 2$ scattering amplitude[1]

$$\lim_{|s|\to\infty} \frac{T_{\text{QFT}}(s,t)}{|s|^2} = 0, \quad t < m_{\text{gap}}^2, \qquad (1.1)$$

where $m_{\text{gap}}^2$ is the location of the first nonanalyticity in $t$ which depends on the details of the theory and the scattering process [1, 2]. The purpose of this paper is to address the question: *What is the bound on the Regge growth of gravitational scattering amplitudes?*

The basic physical principles underlying (1.1) are unitarity, analyticity, and crossing. In addition to that, the textbook derivation of (1.1) heavily relies on the existence of the mass gap and axioms of QFT. The intuitive picture behind (1.1), however, is very simple and reflects upon scattering at large impact parameters. It states that the effective radius of interaction in a gapped theory cannot grow with energy faster than $\log s$. In other words, it follows from the fact that at large impact parameters $b \gtrsim \log s$ interaction between the scatterers decays as $s^N e^{-m_{\text{gap}}b}$,[2] where the Yukawa potential $e^{-m_{\text{gap}}b}$ is an expression of the short-range nature of forces in the gapped theories and the polynomial boundedness $s^N$ is expected on very general grounds of causality [3, 4]. By combining the large impact parameter behavior of the amplitude $s^N e^{-m_{\text{gap}}b}$ with basic physical principles one gets (1.1), see e.g. [5].

---

[1] Here $s$ and $t$ are the usual Mandelstam variables and are respectively the square of the total scattering energy and the momentum transfer. The Regge limit is defined as $|s| \to \infty$ at fixed $t$.

[2] In this paper $A(s) \lesssim B(s)$ stands for $\lim_{|s|\to\infty} \frac{A(s)}{B(s)} < \infty$.

In this paper, we apply the same strategy to gravitational theories, for which high-energy scattering at large impact parameters is famously controlled by known semi-classical physics [6–11]. We combine the basic physical principles of analyticity, subexponentiality, unitarity and crossing with the expected large impact parameter behavior of the amplitude to derive bounds on the Regge limit.

We distinguish three types of bounds on the behavior of the amplitude as a function of $s$:

    a. Pointwise Regge bound for fixed $t < 0$.

    b. Smeared Regge bound for the amplitude integrated over the transferred momentum $t \leq 0$.

    c. Bound on the local growth of the amplitude.

Let us next discuss them in some detail separately.

**a. Pointwise bounds**

The most conservative assumption about the large impact parameter behavior of gravitational scattering is the following: for any given scattering energy $s$, gravitational interactions at large enough impact parameters $b > b_{\text{Born}}(s)$ are accurately captured by a single graviton exchange (or, equivalently, the $t = 0$ pole of the amplitude). Using this, we demonstrate that

$$\text{Born}: \quad |T(s,t)| \lesssim |s|^{2 - \frac{d-7}{2(d-4)}}, \quad t < 0. \quad (1.2)$$

A tighter bound can be derived assuming that the tree-level phase shift eikonalizes [6, 12, 13], and extrapolating the eikonal ansatz for the amplitude up to impact parameters for which inelastic effects become non-negligible. Tidal excitations present the leading inelastic effect in $d > 5$ as shown by Amati, Ciafaloni, and Veneziano (ACV) [7–10]. Assuming that the amplitude for $b > b_{\text{tidal}}(s)$ is accurately captured by the eikonalized tree-level phase shift we get

$$\text{Eikonal} + \text{tidal}\big|_{d>5}: \quad |T(s,t)| \lesssim |s|^{2 - \frac{d-4}{2(d-3)}}, \quad t < 0. \quad (1.3)$$

We expect that the bound (1.3) is saturated by the large impact parameter elastic scattering in any gravitational theory (at least for real $s$).

As demonstrated by ACV, gravitational waves emission is the leading inelastic effect in $d \leq 5$ at high enough energies. Therefore, in $d = 5$ we use the Eikonal+GW model which extrapolates the large impact parameter ansatz for the amplitude given by the eikonalized tree-level phase shift up to impact parameters $b_{\text{GW}}(s)$ for which gravity wave emission effects kick in. In this way we get

$$\text{Eikonal} + \text{GW}\big|_{d=5}: \quad |T(s,t)| \lesssim |s|^{2 - \frac{1}{5}}, \quad t < 0. \quad (1.4)$$

Our results for the pointwise bounds are summarized in Table I. We can summarize them by saying that

| Model | Pointwise | Smeared |
|---|---|---|
| Born | $|s|^{2 - \frac{d-7}{2(d-4)}}$ | $|s|^{2 - \min(1, \frac{\mathsf{a}}{d-4}, \frac{\mathsf{b} + \frac{d-5}{2}}{d-4})}$ |
| Eikonal+tidal$\big|_{d>5}$ | $|s|^{2 - \frac{d-4}{2(d-3)}}$ | $|s|^{2 - \min(1, \frac{\mathsf{a}}{d-4}, \frac{\mathsf{b} + \frac{d-1}{2}}{d-2})}$ |
| Eikonal+GW$\big|_{d=5}$ | $|s|^{2 - \frac{1}{5}}$ | $|s|^{2 - \min(1, \mathsf{a}, \frac{2\mathsf{b}+3}{5})}$ |

TABLE I. Summary of the asymptotic Regge bounds derived in this paper for the elastic $2 \rightarrow 2$ scattering amplitudes in a gravitational theory in $d > 4$. Pointwise bounds refer to $\lim_{|s|\to\infty} |T(s,t)| \lesssim s^{\#}$ for $t < 0$. Smeared bounds refer to the Regge bound $\lim_{|s|\to\infty} |T_{\psi_{\mathsf{a},\mathsf{b}}}(s)| \lesssim s^{\#}$ on the scattering amplitude smeared over the transferred momenta (1.6), where the smearing function $\psi_{\mathsf{a},\mathsf{b}}(q)$ satisfies (1.7). Different models refer to the different large impact parameter ansätze that have been used to estimate the amplitude. Using the results above, we see that the amplitude satisfies (1.9). Taking into account black hole formation further improves the bound to (1.10).

the scattering amplitude for fixed $t < 0$ admits *twice-subtracted dispersion relations* (2SDR)

$$\lim_{|s|\to\infty} \frac{T(s,t)}{|s|^2} = 0. \quad (1.5)$$

Note that in the most conservative Born model, (1.5) holds only for $d > 7$; three subtractions are required in $d = 6, 7$; in $d = 5$ four subtractions are needed.

**b. Smeared bounds**

We also consider a smeared scattering amplitude [14, 15]

$$T_{\psi_{\mathsf{a},\mathsf{b}}}(s) \equiv \int_0^{q_0} dq\, q\, [\psi_{\mathsf{a},\mathsf{b}}(q) T(s, -q^2)], \quad (1.6)$$

where we defined the momentum transfer $\vec{q}$ via $t = -\vec{q}^2$ and wrote its norm as $|\vec{q}| = q$; $\mathsf{a}$ and $\mathsf{b}$ refer to the behavior of $\psi_{\mathsf{a},\mathsf{b}}(q)$ close to the end-points[3][4]

$$\psi_{\mathsf{a},\mathsf{b}}(q) \overset{q\to 0}{\sim} q^{\mathsf{a}}, \quad \mathsf{a} > 0,$$
$$\psi_{\mathsf{a},\mathsf{b}}(q) \overset{q\to q_0}{\sim} (q_0 - q)^{\mathsf{b}}, \quad \mathsf{b} \geq 0, \quad (1.7)$$

where in the paper we will restrict considerations to $\mathsf{b} \geq 0$ only. This could be in principle relaxed to

————

[3]The condition $\mathsf{a} > 0$ is necessary for convergence of the smeared amplitude due to the presence of the massless pole $\sim \frac{1}{q^2}$. In the gapped theories one can consider $\mathsf{a} > -2$.

[4]We use the following notation: $A \overset{x\to x_0}{\sim} B$ stands for $\lim_{x\to x_0} \frac{A}{B} = $ const.

b $> -1$. Furthermore, we require the function $\psi_{\mathsf{a},\mathsf{b}}(q)$ to be $C^{\min(\mathsf{a},\mathsf{b})}$.

We show that the smeared amplitude satisfies bounds presented in Table I. A new step compared to the pointwise analysis is that to derive the bounds we

   i) establish the validity of dispersion relations with a few subtractions.

   ii) use dispersion relations to improve some of our estimates.

The results of Table I can be succinctly summarized by saying that the smeared scattering amplitude admits 2SDR[5]

$$\lim_{|s|\to\infty} \frac{T_{\psi_{\mathsf{a},\mathsf{b}}}(s)}{|s|^2} = 0. \tag{1.8}$$

This conclusion agrees with the AdS/CFT based analysis of [16], where the rigorous AdS bounds derived from the CFT correlators reduced to the bounds from 2SDR in the flat space limit. This fact can be taken as an indirect evidence for nonperturbative validity of our assumptions.

In particular, we see from the table that, independently of the model, we get

$$|T_{\psi_{\mathsf{a}\geq d-4, \mathsf{b}\geq \frac{d-3}{2}}}(s)| \lesssim |s|. \tag{1.9}$$

The effect of extra conditions on the smearing function is to suppress scattering at large impact parameters.[6] This effect is purely kinematical and relies on the properties of the $d$-dimensional Legendre polynomials.

Finally, by combining 2SDR with the classical picture of black hole formation at large energies and fixed impact parameters we arrive at a stronger bound

$$\text{BH}: \quad \lim_{|s|\to\infty} \frac{T_{\psi_{\mathsf{a}> d-4, \mathsf{b}> \frac{d-3}{2}}}(s)}{|s|} = 0. \tag{1.10}$$

In other words, the number of subtractions in this case is one. The bound (1.10) also holds in holographic QCD [18, 19], and it is conceivable that it applies to standard QCD as well [20–22].

### c. Local growth bounds

Finally, we analyze bounds on the local growth of the amplitude [4].[7] By appropriately choosing the smearing function $\psi_{\mathsf{a},\mathsf{b}}(q)$ and using 2SDR we establish a version of the CRG conjecture [25] that bounds the local growth

of the scattering amplitude of scalar particles in a weakly coupled gravitational theory by $|s|^2$ for physical $-M_{\text{gap}}^2 \leq t \leq 0$.[8]

Similarly, we show that in the gapped theories the local growths of the amplitude at fixed $t$ is bounded by $s^2$ for $0 \leq t < m_{\text{gap}}^2$.[9] When applied to the planar gauge theories, e.g. the large $N$ QCD, this result is an extension of the CRG conjecture to the gapped theories.

$$* \quad * \quad *$$

The most interesting case of gravitational scattering in $d = 4$ requires a more careful treatment of the asymptotic states [26–29] to define the scattering amplitude which is beyond the scope of the present paper. It would be very interesting to generalize our analysis to the properly defined IR-finite scattering observables in four dimensions. We comment on this case more extensively in the conclusions.

## II. ASSUMPTIONS

We consider a relativistic, gravitational theory in $d \geq 5$. By this we simply mean that the spectrum of the theory contains a massless particle of spin two.[10] Our assumptions are the following:

   0. **Unitarity**: Dynamics is described by a unitarity $S$-matrix which characterizes transitions between asymptotic states [30]. We take the asymptotic states to be given by a set of free, stable particles.

For simplicity we focus on the elastic two-to-two scattering amplitude

$$A, B \to A, B \quad , \tag{2.1}$$

and we take external states to be scalar particles of mass $m$. Generalization to the spinning case should be more or less direct, but we do not perform it here. The scattering process is characterized by a function of two variables $T(s,t) \equiv T^{A,B\to A,B}(s,t)$.

   1. **Analyticity**: Given fixed $t < 0$, there exists $s_0(t)$ such that $T(s,t)$ is analytic in the upper half-plane for $|s| > s_0(t)$. This assumption is associated with causality.[11] We assume the amplitude in the lower

---

[5]In the most conservative Born model and $d = 5$, it requires that b $> 0$.

[6]This point has been recently emphasized in [17].

[7]In the context of AdS/CFT the corresponding bound is related to the bound on chaos, see appendix A in [23]. The relation of the bound on chaos to the local growth of the flat space amplitudes was recently explored in [24].

[8]Here $M_{\text{gap}}^2$ is the gap in the spectrum of stringy states (or, more generally, states that contribute at leading order in the gravitational coupling).

[9]This is the same as saying that the leading Regge trajectory $j(t) \leq 2$ for $t < m_{\text{gap}}^2$.

[10]For example, the theory can also contain photons and other stable particles. This does not affect our analysis.

[11]Perturbatively one can construct unitary $S$-matrices that slightly violate causality [31–33]. They have singularities in the upper half-plane and therefore violate the analyticity assumption that we make here. Whether such theories are nonperturbatively well-defined is not clear [34].

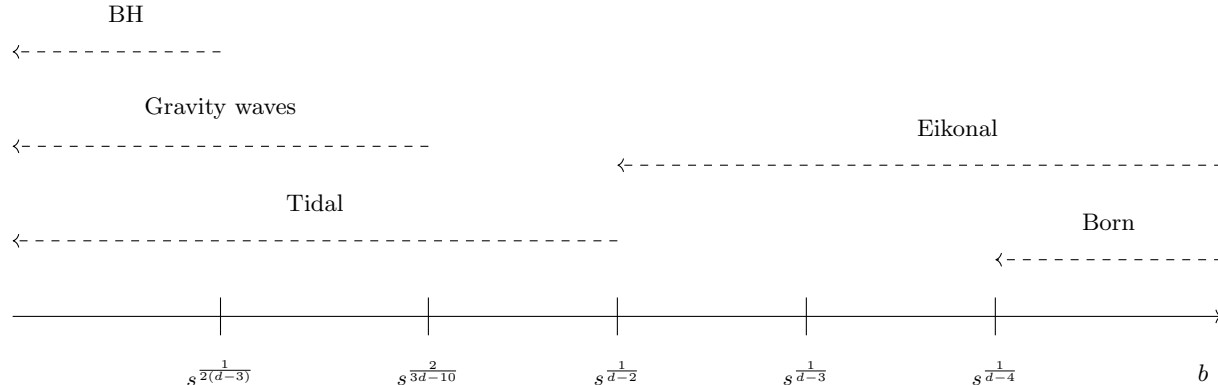

FIG. 1. Different energy scales in the impact parameter space $b$. The scales are arranged as they appear in the large $s$ limit in $d \geq 7$. The Born approximation to the gravitational interaction is valid up to impact parameters $b \gtrsim s^{\frac{1}{d-4}}$. For impact parameters $s^{\frac{1}{d-2}} \lesssim b \lesssim s^{\frac{1}{d-4}}$ gravitational interaction is controlled by an eikonalized Born result. Moreover in the Regge limit, the leading contribution to the scattering amplitude comes from impact parameters $\sim s^{\frac{1}{d-3}}$, see e.g. [11]. For $b \lesssim s^{\frac{1}{d-2}}$ tidal effects become important. Similarly, for $b \lesssim s^{\frac{2}{3d-10}}$ emission of gravitational waves becomes relevant. Finally, for impact parameters less than the corresponding Schwarzschild radius $b \lesssim s^{\frac{1}{2(d-3)}}$ scattering is dominated by the black hole formation. For $d \geq 7$ the leading inelastic effect is due to tidal excitations; for $d = 6$ tidal exciatations and gravity wave emission scale with $s$ in the same way; finally, for $d \leq 5$ the leading inelastic effect at large energies is due to the emission of gravity waves.

half-plane to be related to the upper half-plane by hermitian analyticity

$$T(s^*, t) = T^*(s, t). \qquad (2.2)$$

2. **Subexponentiality**: Given fixed $t < 0$, we assume that the amplitude $T(s, t)$ is bounded as

$$|T(s, t)| \leq e^{C|s|^\beta}, \quad \beta < 1, \qquad (2.3)$$

everywhere in the region of analyticity in the upper half-plane $\arg(s) \in (0, \pi)$. This condition is related to asymptotic causality [4, 35], but to the best of our knowledge it has not been rigorously proven starting from it, therefore we list it as an assumption.[12]

3. **Crossing**: We assume that the $u$-channel cut describes scattering in the crossed channel

$$T_{A,B \to A,B}(u + i\epsilon, t) = T^*_{A,\bar{B} \to A,\bar{B}}(s + i\epsilon, t), \qquad (2.4)$$

where $\bar{B}$ stands for an anti-particle and $s + t + u = 4m^2$. Given our assumption of analyticity the $s$- and $u$-channel are connected via analytic continuation in the upper half-plane. In other words, we assume crossing symmetry, see [37] for a recent discussion.

4. **Regularity**: We treat the scattering amplitude $T(s, t)$ for physical $s$ and $t$ as a function and not as a distribution which it strictly speaking is [38]. We expect that this does not invalidate any of our bounds (as long as they are understood on average in $s$).

In the gapped theories, analyticity has been proven in an extended region that includes positive $t$ and does not depend on $s$ [1]. Extended analyticity together with the properties 0-4 are enough to prove (1.1). In a gravitational theory, we do not have this extra analyticity region and the basic assumptions reviewed above are not sufficient to derive the Regge bound. An extra ingredient that allows us to make progress is an assumption about the large $J$ behavior of partial waves. It concerns scattering at large impact parameters and can be formulated as follows[13]

5. **Gravitational clustering**: For any fixed physical $s$, there exists impact parameter $b_*(s)$, such that scattering for $b > b_*(s)$ is well-described by general relativity. Via a familiar relation $b \sim \frac{2J}{\sqrt{s}}$ it is the statement about the behavior of partial waves at large spin $J$. Finally, via the Fourier transform it is the statement about the behavior of the scattering amplitude close to $t = 0$.

We will elaborate on the precise form of $b_*(s)$ below when deriving the bounds. Let us point out that the

---

[12] In axiomatic QFT, a stronger condition of polynomial boundedness is eventually attributed to temperedness of correlation functions of local operators from which the scattering amplitudes are derived via the LSZ procedure, see e.g. [3]. It can be also established in the Haag-Kastler axiomatic framework [36].

[13] The fact that every relativistic theory with a massless spin two particle in its spectrum reproduces general relativity at large distances is originally due to Weinberg [39, 40].

notion of gravitational clustering is different from the usual clustering which states that interactions go to zero as $b \to \infty$. For example, in $d = 4$ gravitational clustering holds, whereas clustering breaks down.

Assumptions $0 - 4$ are true in the context of nonperturbative QFT and perturbative string theory [41]. Assumption 5 is based on the detailed analysis of high energy gravitational scattering [6–10, 12, 13], see also [42, 43]. To set the stage for our analysis, we review the relevant energy scales in Figure 1.[14]

### A. Strategy

We would like to constrain the behavior of the amplitude at large $|s|$. By assumption the amplitude is analytic outside of the region marked by a blue line in figure 2. Our strategy is then simple: we use unitarity, crossing and gravitational clustering to constrain the growth of the amplitude along the real axes, regions (a) and (b). Together with the assumptions of analyticity and subexponentiality in the upper half-plane, region (c), this implies polynomial boundedness in the upper half-plane via the Phragmén-Lindelöf principle.

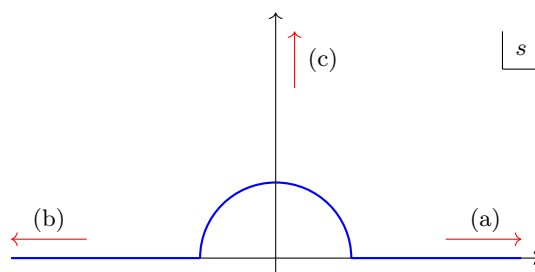

FIG. 2. The upper half-plane in the complex $s$ variable. In this paper we mostly concern ourselves with bounding the scattering amplitude $T(s, t)$ along the real axes (regions (a) and (b)). We then combine this with analyticity in the upper half-plane together with the subexponentiality assumption in region (c) to derive the Regge bounds on the amplitude everywhere in the upper half-plane.

### III. PARTIAL WAVE EXPANSION

Nonperturbative unitarity will play a key role in our derivation of the Regge bounds. It is particularly easy to state by expanding the amplitude in a complete set of partial waves introduced in this section.

As a preparation for this step, let us briefly recall the behavior of the scattering amplitude $T(s, t)$ for physical $t$. Every elastic scattering amplitude in a gravitational theory is singular at $t = 0$.[15] This is due to the universal nature of gravitational attraction which translates to the fact that any pair of particles can exchange gravitons, see Figure 3.

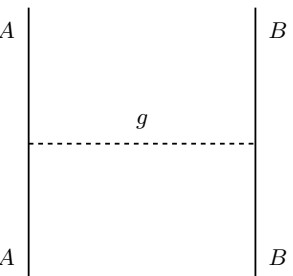

FIG. 3. Every elastic amplitude in a gravitational theory is singular at $t = 0$ due to universal, long-range nature of gravity.

At tree-level the amplitude takes the form[16]

$$T_{\text{tree}}(s, t) = -\frac{8\pi G_N s^2}{t} \ . \tag{3.1}$$

Translated to the impact parameter space,[17] the $\frac{1}{t}$ pole in (3.1) leads to long-range interaction between particles which decays as $\frac{1}{b^{d-4}}$.

In the quantum theory the pole becomes a branch-point, nevertheless the leading $t \to 0$ asymptotic at fixed $s$ remains unchanged [8, 10, 11]

$$T(s, t) = -\frac{8\pi G_N s^2}{t} \left(1 + O(G_N s |t|^{\frac{d-4}{2}})\right). \tag{3.2}$$

Formula (3.2) is an expression of gravitational clustering.[18] The correction $O(G_N s |t|^{\frac{d-4}{2}})$ was derived in [8, 10, 11] using the eikonalized tree-level exchange. Its precise form is not important to us at the moment. The crucial point is that for any fixed $s$ there is a region around $t = 0$ where the amplitude is controlled by the graviton pole.

Note that as $s$ becomes large the region in which the tree-level approximation is valid shrinks to zero. This is directly related to the fact that in the impact parameter space the tree-level amplitude captures scattering at

---

[14] The corresponding picture in AdS has different scales all proportional to $\log s$ (similar to what happens in the gapped theories in flat space), see [44, 45]. This is related to the fact that on super-AdS scale gravitational interaction is controlled by the Yukawa potential.

---

[15] For scattering of identical particles the same is true for $s, u = 0$.

[16] Since we will be interested in the large $s$ limit, we drop the $\frac{m^2}{s}$ corrections. The full amplitude for the massive case can be found, for example, in [13].

[17] Or, equivalently, doing the Fourier transform with respect to $\vec{q}$, where $t = -\vec{q}^2$, see (3.19).

[18] The breakdown of the leading singularity being a pole in $d = 4$ is closely related to the fact that gravitons are good asymptotic states in $d > 4$ and not in $d = 4$.

impact parameters $b > b_{\text{Born}}(s)$ where $b_{\text{Born}}(s)$ grows with $s$.

After this small digression we are now ready to expand the amplitude in a complete set of partial waves

$$T(s,t) = \frac{1}{2} \sum_{J=0}^{\infty} n_J^{(d)} f_J(s) P_J^{(d)}\left(1 + \frac{2t}{s - 4m^2}\right), \quad (3.3)$$

where $P_J^{(d)}(\cos\theta)$ are the orthogonal $d$-dimensional Legendre (or Gegenbauer) polynomials. Our conventions are summarized in Appendix A, and we define $\tilde{n}_J^{(d)} \equiv n_J^{(d)}/2$ to simplify the notation. The partial waves are computed as follows

$$f_J(s) = \mathcal{N}_d \int_{-1}^{1} dz \, (1 - z^2)^{\frac{d-4}{2}} P_J^{(d)}(z) \, T(s, t(z)) \,, \quad (3.4)$$

where $z \equiv \cos\theta$ is the scattering angle.[19]

Let us discuss the convergence properties of the partial wave expansion of the scattering amplitudes that satisfy (3.2), which we understand as the limit of partial sums

$$\sum_{J=0}^{\infty} \tilde{n}_J^{(d)} f_J(s) P_J^{(d)}(z) \equiv \lim_{J_{\text{max}} \to \infty} \sum_{J=0}^{J_{\text{max}}} \tilde{n}_J^{(d)} f_J(s) P_J^{(d)}(z). \quad (3.5)$$

As we review in Appendix B, it readily follows from (3.2) that the partial wave expansion converges pointwise for $-1 < \cos\theta < 1$ in $d > 5$, and converges as a distribution in $d = 5$.[20]

A simple way to understand the convergence of the partial wave expansion is to inspect the large $J$ behavior of the sum (3.3). As $J$ becomes large the partial wave integral (3.4) effectively localizes close to $t = 0$ due to the oscillatory nature of the Legendre polynomials and therefore the small $t$ expansion (3.2) translates to the large $J$ expansion.[21] In this way we obtain[22]

$$f_J(s) \simeq \frac{\Gamma\left(\frac{d-4}{2}\right)}{(4\pi)^{\frac{d-4}{2}}} \frac{G_N s}{J^{d-4}}, \quad J \to \infty. \quad (3.6)$$

Using that $\left| P_J^{(d)}(\cos\theta) \right| \sim J^{\frac{3-d}{2}}$ at fixed angle $-1 < \cos\theta < 1$, and the fact that $n_J^{(d)} \sim J^{d-3}$ we conclude that the partial wave expansion converges absolutely for $d > 7$. In $d = 6, 7$ partial sums (3.5) converge thanks to the oscillatory nature of the Legendre polynomials.

————

[19] In writing (3.3) and (3.4) we took into account the fact that $A$ and $B$ are nonidentical particles.

[20] The situation in $d = 5$ is analogous to the simple example discussed in [46].

[21] In the gapped theories the situation is exactly the same [47]. Expansion around the nearest nonanalyticity (here expansion around $t = 0$) controls the large $J$ behavior of the partial waves.

[22] Here $A \simeq B$ for $x \to \infty$ means that $\lim_{x\to\infty} \frac{A}{B} = 1$.

## A. Smeared partial waves

We will be also interested in the high-energy behavior of the amplitude smeared over the transferred momentum

$$T_{\psi_{\text{a,b}}}(s) \equiv \int_0^{q_0} dq \, q[\psi_{\text{a,b}}(q) T(s, -q^2)]. \quad (3.7)$$

Due to the singularity of $T(s, -q^2)$ at $q^2 = 0$ the integral (3.7) is only well-defined if

$$\psi_{\text{a,b}}(q) \overset{q \to 0}{\sim} q^{\text{a}}, \quad a > 0. \quad (3.8)$$

It is possible to decompose the smeared amplitude into partial waves as follows

$$T_{\psi_{\text{a,b}}}(s) = \sum_{J=0}^{\infty} \tilde{n}_J^{(d)} f_J(s) P_J^{(d)}[\psi_{\text{a,b}}] \,, \quad (3.9)$$

where we defined

$$P_J^{(d)}[\psi_{\text{a,b}}] \equiv \int_0^{q_0} dq \, q \left[ \psi_{\text{a,b}}(q) P_J^{(d)}\left(1 - \frac{2q^2}{s - 4m^2}\right) \right] \,. \quad (3.10)$$

In (3.9) we permuted the action of the $\psi_{\text{a,b}}$-functional and the sum over partial waves. This is a nonobvious step, analogous to the swapping property discussed in [48], and we establish it in Appendix C.

As opposed to the pointwise case, the smeared partial wave expansion converges also in $d = 5$. It encapsulates the fact that the pointwise partial wave expansion converges as a distribution.

## B. Unitarity

Nonperturbative unitarity can be conveniently expressed at the level of the partial waves $f_J(s)$. In order to do so, we introduce the following combination [49]

$$S_J(s) \equiv 1 + i \frac{(s - 4m^2)^{\frac{d-3}{2}}}{\sqrt{s}} f_J(s). \quad (3.11)$$

Unitarity states that

$$|S_J(s)| \leq 1, \quad s \geq 4m^2. \quad (3.12)$$

Equivalently, we can rewrite it as follows

$$2\text{Im} f_J(s) \geq \frac{(s - 4m^2)^{\frac{d-3}{2}}}{\sqrt{s}} |f_J(s)|^2 \,. \quad (3.13)$$

This can be solved in terms of the phase shifts $\delta_J(s)$

$$f_J(s) = \frac{\sqrt{s}}{(s - 4m^2)^{\frac{d-3}{2}}} i(1 - e^{2i\delta_J(s)}) \,, \quad (3.14)$$

with the unitarity constraint being $\text{Im}[\delta_J(s)] \geq 0$.

The condition (3.13) does not depend on the convergence properties of the partial wave expansion and holds whenever the scattering amplitude is well-defined, namely for $d \geq 5$.

## C. Impact parameter representation

It is instructive to discuss the impact parameter representation of the amplitude, which describes scattering of particles at large energies separated by some fixed distance $b$ in the transverse direction.

Before deriving it, let us first make a connection with the classical picture of two particles with equal but opposite momenta $\pm \vec{p}$ separated by distance $b$ in the transverse space, see Figure 4.

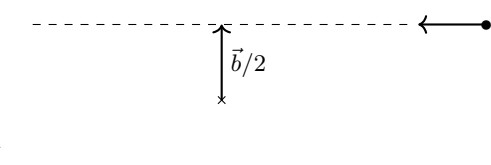

FIG. 4. The classical picture of scattering at fixed impact parameter $b$.

Using the familiar formula $\vec{J} = \vec{b} \times \vec{p}$ we get for the angular momentum $J = \frac{b\sqrt{s-4m^2}}{2}$. We are interested in the high energy behavior of the amplitude. We therefore consider the double-scaling limit $J \to \infty$, $s \to \infty$ with the ratio

$$b \equiv \frac{2J}{\sqrt{s}} \tag{3.15}$$

held fixed.

We take this limit at the level of the partial wave expansion (3.3). It is easy to check that in this limit we have

$$\frac{n_J^{(d)} P_J^{(d)}(1 - \frac{q^2 b^2}{2J^2})}{J^{d-3}} \overset{J \to \infty}{\simeq} 2^d (2\pi)^{\frac{d-2}{2}} (bq)^{\frac{4-d}{2}} J_{\frac{d-4}{2}}(bq) . \tag{3.16}$$

Let us then assume that the phase shift in (3.14) admits the limit as well

$$\delta_{J = \frac{b\sqrt{s}}{2}}(s) = \delta(s, b) + \dots , \quad s \to \infty, \tag{3.17}$$

where $\delta_J(s)$ was defined in (3.14) and ... are suppressed in the large $s$ limit. We expect that this limit is well-defined at least in the semi-classical regime where it describes scattering at fixed impact parameter $b$, see e.g. [7–10, 50, 51].

Unitarity of the $S$-matrix (3.12) then becomes

$$|e^{2i\delta(s,b)}| \leq 1 + \dots , \quad b \neq 0, \tag{3.18}$$

where ... stand for the terms that decay in the large $s$ limit.

Switching in (3.14) from $\delta_J(s)$ to the impact parameter phase shift $\delta(s, b)$ and switching from the sum to the integral $\sum_J \to \frac{\sqrt{s}}{2} \int db$ we arrive at the impact parameter representation for the amplitude at large $s$

$$T(s,t) \simeq 2is(2\pi)^{\frac{d-2}{2}} \int_0^\infty db\, b^{d-3}(bq)^{\frac{4-d}{2}} J_{\frac{d-4}{2}}(bq)$$

$$\times \left(1 - e^{2i\delta(s,b)}\right) = 2is \int d^{d-2}\vec{b}\, e^{-i\vec{q}\vec{b}} \left(1 - e^{2i\delta(s,|\vec{b}|)}\right), \tag{3.19}$$

where in the last line, we reintroduced vector $\vec{b}$ to highlight the statement that the impact parameter representation is the Fourier transform of the scattering amplitude. By construction (3.19) has corrections which are suppressed by a power of $\frac{1}{sb^2}$.

Below we will mostly work with the partial wave expansion and we will only use the impact parameter representation in the semi-classical regime. We find that the impact parameter representation is a very illuminating way of thinking about different $J$'s that contribute to the partial wave sum at large $s$, see Figure 1.

## IV. POINTWISE REGGE BOUNDS

In what follows, we consider $s$ to be large, real and positive (i.e. (a) direction in Figure 2). We will be only interested in scaling of various quantities in the large $s$ limit. We start by splitting the sum over $J$'s into two parts (low-spin and high-spin)

$$T(s,t) = T^{\leq J_*}(s,t) + T^{>J_*}(s,t), \tag{4.1}$$

$$T^{\leq J_*}(s,t) \equiv \sum_{J=0}^{J_*} \tilde{n}_J^{(d)} f_J(s) P_J^{(d)}\left(1 - \frac{2q^2}{s - 4m^2}\right) , \tag{4.2}$$

$$T^{>J_*}(s,t) \equiv \sum_{J=J_*+1}^{\infty} \tilde{n}_J^{(d)} f_J(s) P_J^{(d)}\left(1 - \frac{2q^2}{s - 4m^2}\right) . \tag{4.3}$$

At this point $J_*$ is arbitrary and we will fix it later to optimize the bound on $|T(s,t)|$.

To bound the low-spin contribution we simply use unitarity

$$|f_J(s)| \lesssim s^{2-\frac{d}{2}} , \tag{4.4}$$

and the following lemma

**Lemma 1:** The $d$-dimensional Legendre polynomials satisfy

$$\left| P_J^{(d)}\left(1 - \frac{2q^2}{s}\right)\right| \leq \min\left(1, C(d)\left(\frac{Jq}{\sqrt{s}}\right)^{\frac{3-d}{2}}\right), \tag{4.5}$$

for $d \geq 3$. For $d < 5$ the inequality (4.5) has been rigorously proven, see [52] Theorem 7.33.2. We use it for $d \geq 5$ based on the large $J$ analysis of $P_J^{(d)}$ and numerical experimentation.

For spins $J \lesssim \sqrt{s}$, (4.5) bounds the Legendre polynomials by 1, whereas for $J \gtrsim \sqrt{s}$ the $J$-dependent bound becomes optimal. We then get the following estimate

$$|T^{\leq J_*}(s,t)| \leq \sum_{J=0}^{J_*} \tilde{n}_J^{(d)} |f_J(s)| \left| P_J^{(d)} \left( 1 - \frac{2q^2}{s - 4m^2} \right) \right|$$

$$\leq s^{2-\frac{d}{2}} \sum_{J=0}^{J_*} \tilde{n}_J^{(d)} \left| P_J^{(d)} \left( 1 - \frac{2q^2}{s - 4m^2} \right) \right|$$

$$\lesssim s^{\frac{5-d}{4}} J_*^{\frac{d-1}{2}} + O(s), \qquad (4.6)$$

where $O(s)$ comes from the contribution of spins $J \lesssim \sqrt{s}$, whereas the spin-dependent bound comes from $J \gtrsim \sqrt{s}$.

Next, we consider the high-spin part $T^{>J_*}(s,t)$ and bound it using different models of the large impact parameter scattering.

### A. The Born model

To analyze the high-spin contribution, we first make a conservative assumption and introduce $J_{\text{Born}}(s,\epsilon)$, the spin above which the tree-level approximation to the partial waves becomes accurate. It is defined by imposing that the tree-level phase shift is small

$$\delta_J^{\text{tree}} = \frac{\Gamma\left(\frac{d-4}{2}\right)}{(4\pi)^{\frac{d-4}{2}}} \frac{G_N s^{\frac{d-2}{2}}}{J_{\text{Born}}^{d-4}} = \epsilon \ll 1, \qquad (4.7)$$

where $\epsilon$ is some small fixed number. From the equation above, we see that $J_{\text{Born}} \sim s^{\frac{d-2}{2(d-4)}}$.

We are now ready to state the weakest form of our gravitational clustering assumption: there exists $\epsilon$ and $C_0$ such that for arbitrarily large $s$ we have

$$J > J_{\text{Born}}(s,\epsilon): \quad |f_J(s)| \leq C_0 G_N s J^{4-d}. \qquad (4.8)$$

This is the statement that no matter how big the collision energy is at large enough impact parameters gravity becomes weakly coupled and is well-approximated by the tree-level result.

In this way we get for the high-spin contribution, assuming $J_* \geq J_{\text{Born}}(s,\epsilon)$,

$$|T^{>J_*}(s,t)| \leq \sum_{J=J_*+1}^{\infty} \tilde{n}_J^{(d)} |f_J(s)| \left| P_J^{(d)} \left( 1 - \frac{2q^2}{s - 4m^2} \right) \right|$$

$$\lesssim s^{\frac{d+1}{4}} J_*^{\frac{7-d}{2}}, \qquad (4.9)$$

where the sum only converges for $d > 7$.

Combining the two bounds we get

$$|T(s,t)| \leq |T^{\leq J_*}(s,t)| + |T^{>J_*}(s,t)|$$

$$\lesssim \min_{J_* \geq J_{\text{Born}}(s,\epsilon)} \left\{ s^{\frac{5-d}{4}} J_*^{\frac{d-1}{2}}, s^{\frac{d+1}{4}} J_*^{\frac{7-d}{2}} \right\}$$

$$\lesssim s^{2-\frac{d-7}{2(d-4)}}, \quad d > 7. \qquad (4.10)$$

where the optimal spin is $J_* \sim J_{\text{Born}}(s,\epsilon) \sim s^{\frac{d-2}{2(d-4)}}$. For this choice of $J_*$, the contribution of both low spin and high spin partial waves are of the same order. The condition $d > 7$ is the same condition that appeared in our discussion of the absolute convergence of the partial wave expansions.

The estimate above can be improved if instead of the bound (4.8), we impose that for $J > J_{\text{Born}}(s,\epsilon)$ the partial waves are controlled (and not just bounded) by the tree-level result up to small corrections

$$J > J_{\text{Born}}(s,\epsilon): \quad f_J(s) \simeq \frac{\Gamma\left(\frac{d-4}{2}\right)}{(4\pi)^{\frac{d-4}{2}}} \frac{G_N s}{J^{d-4}}. \qquad (4.11)$$

This allows us to slightly improve our large $s$ estimate of the high spin contribution

$$T^{>J_*}(s,t) \simeq \sum_{J=J_*+1}^{\infty} \tilde{n}_J^{(d)} G_N s J^{4-d} P_J^{(d)} \left( 1 - \frac{2q^2}{s - 4m^2} \right)$$

$$\lesssim s^{\frac{d+1}{4}} J_*^{\frac{5-d}{2}}, \quad d > 5, \qquad (4.12)$$

where $d > 5$ is precisely the condition that partial wave expansion converges pointwise. Finally, in $d = 5$ we can explicitly subtract $T_{\text{tree}}(s,t)$ from $T^{>J_*}(s,t)$ and estimate the difference in the same way as above.

To summarize, using the Born model, we conclude that for $s$ large and real

$$\lim_{s \to \infty} |T(s,t)| \lesssim s^{2 - \frac{d-7}{2(d-4)}}. \qquad (4.13)$$

Using crossing (2.4) we can apply the same analysis to the $u$-channel cut (i.e. direction (b) in Figure 2). Finally in the upper half-plane, direction (c) in Figure 2, we assume subexponentiality (2.3). As a result, we can apply the maximum modulus principle to the upper half-plane (or the Phragmén-Lindelöf principle) to conclude that

$$\lim_{|s| \to \infty} |T(s,t)| \lesssim |s|^{2 - \frac{d-7}{2(d-4)}}. \qquad (4.14)$$

In particular, for $d > 7$ this implies that we can write dispersion relations with two subtractions

$$\lim_{|s| \to \infty} \frac{T(s,t)}{|s|^2} = 0, \quad t < 0, \quad d > 7. \qquad (4.15)$$

In $d = 6, 7$, we can write dispersion relations with three subtractions, and in $d = 5$, four subtractions are needed.

### B. The Eikonal model

Some readers might think that we were overcautious with our estimates and a better bound should exist. To analyze this question, we consider next an extension of the Born model.

The basic idea is that as we go to the impact parameters smaller than $b < b_{\text{Born}}(s, \epsilon)$ the first thing that happens is that the tree-level phase shift exponentiates or eikonalizes, see Figure 1. In this regime, the amplitude is purely elastic even though the phase shift $\delta_{\text{tree}}(s, b)$ is large, and the scattering amplitude is captured by propagation of a particle on the Aichelburg-Sexl shockwave background created by another particle [6, 12, 13]. A natural question is: down to which impact parameters the eikonalization of the tree-level phase shift is an accurate approximation? Our working assumption is that it holds up to the point where the first inelastic effect becomes tangible.

In $d > 5$, leading inelasticity comes from the tidal effects [8, 10]. The relevant impact parameter $b_{\text{tidal}}(s, \epsilon)$ is defined such that the imaginary part of the phase shift is approximately equal to

$$\text{Im}\,\delta(s, b_{\text{tidal}}) \approx \frac{G_N s}{b_{\text{tidal}}^{d-2}} \frac{1}{m_{\text{tidal}}^2} = \epsilon \ll 1 \,, \qquad (4.16)$$

where $m_{\text{tidal}}$ is a characteristic scale of the internal excitations of the scattered particle, e.g. the string scale, and $\epsilon$ is an arbitrary constant which we take to be small and fixed. Note that $b_{\text{tidal}}(s, \epsilon) \sim s^{\frac{1}{d-2}}$. In $d \leq 5$ the leading inelastic effect at large energies is due to the emission of gravity waves [8, 10]. The corresponding impact parameters at which the effect becomes important scale with energy as $b_{\text{GW}}(s, \epsilon) \sim s^{\frac{2}{3d-10}}$.

Let us next introduce the Eikonal+tidal model used to estimate the high-spin part of the partial wave expansion. For impact parameters $b \geq b_{\text{tidal}}(s, \epsilon)$, there exists $\epsilon$ for which we can reliably estimate the amplitude using the eikonal ansatz (3.19) with [23]

$$b \geq b_{\text{tidal}}(s, \epsilon): \qquad \delta(s, b) = \delta_{\text{tree}}(s, b) = \frac{\Gamma\left(\frac{d-4}{2}\right)}{(\pi)^{\frac{d-4}{2}}} \frac{G_N s}{b^{d-4}}. \qquad (4.17)$$

Note that we do not require the presence of the tidal excitations in the scattering amplitude. We are simply making an assumption about the amplitude at large impact parameters. In $d = 5$, the Eikonal+GW model uses the ansatz (4.17) for $b \geq b_{\text{GW}}(s, \epsilon)$.

Let us proceed with the estimate of the low-spin contribution to the amplitude in the Eikonal+tidal model. The estimate is given by the same formula (4.6) with the only difference being that we choose $J_* \sim J_{\text{tidal}}(s, \epsilon) \sim s^{\frac{d}{2(d-2)}}$ according to (4.16), which gives

$$|T^{\leq J_{\text{tidal}}}(s, t)| \lesssim s^{2 - \frac{d-3}{2(d-2)}}. \qquad (4.18)$$

To estimate the high-spin contribution we have to estimate the following integral

$$T^{>J_{\text{tidal}}}(s, t) \simeq 2is(2\pi)^{\frac{d-2}{2}}$$
$$\int_{b_{\text{tidal}}(s, \epsilon)}^{\infty} db\, b^{d-3}(bq)^{\frac{4-d}{2}} J_{\frac{d-4}{2}}(bq)\left(1 - e^{2i\delta_{\text{tree}}(s, b)}\right). \qquad (4.19)$$

A similar integral was analyzed in detail in [8, 11] with the difference that the integral over $b$ went from 0 to $\infty$.[24] As in [8, 11], we find that the value of the integral (4.19) is controlled by the saddle point located at $b_{\text{eik}} \sim s^{\frac{1}{d-3}}$ and introducing the lower limit at $b_{\text{tidal}}(s, \epsilon) \sim s^{\frac{1}{d-2}}$ does not affect the integral.

The final result is that we have the following high-spin estimate

$$|T^{>J_{\text{tidal}}}(s, t)| \sim s^{2 - \frac{(d-4)}{2(d-3)}}. \qquad (4.20)$$

Combining (4.20) with the low-spin estimate (4.18), we get the following bound on the amplitude

$$\lim_{s \to \infty} |T(s, t)| \lesssim s^{2 - \frac{d-4}{2(d-3)}}\,. \qquad (4.21)$$

Because this bound is saturated by the eikonal amplitude that describes scattering at large impact parameters we believe that it is optimal and is saturated in gravitational theories.

In $d = 5$ the low-spin contribution is estimated as

$$|T^{\leq J_{\text{GW}}}(s, t)| \lesssim s^{2 - \frac{1}{5}}. \qquad (4.22)$$

For the high-spin contribution, the integral analogous to (4.19) (with the lower limit being $b_{\text{GW}}(s, \epsilon)$) is divergent, which is a reflection of the divergence of the partial wave expansion. As in the Born model, we can add and subtract the tree-level result to estimate the integral. The conclusion is that the saddle point expression derived in [8, 11] is a correct estimate of the high-spin part in $d = 5$ and it grows slower than (4.22).

Running the same argument in the $u$-channel and using subexponentiality (2.3), we conclude that (4.21) and (4.22) hold in fact for $|s| \to \infty$. In particular, we see that the amplitude satisfies

$$\lim_{|s| \to \infty} \frac{T(s, t)}{|s|^2} = 0, \quad t < 0\,, \qquad (4.23)$$

which implies that we can write twice-subtracted dispersion relations.

How robust is the model above towards adding small corrections to the phase shift $\delta(s, b)$? Based on the ACV analysis and the fact that $\epsilon$ can be taken arbitrarily small we expect that all such corrections stay small everywhere

---

[23] We will later check that adding small corrections to this model does not change the conclusions. In particular, this implies that writing $\delta(s, b) \simeq \delta_{\text{tree}}$ in (4.17) does not change the result.

[24] See formula (11) in [11], or formula (6.18) in [8].

along the integration region (4.19) and therefore neither affect the position of the saddle, nor the value of the integral. In fact, we see that we could have chosen a cut-off in the impact parameter higher than $b_{\text{tidal}}(s, \epsilon)$ (so that the low-spin and the high-spin contribution are of the same order), which would slightly relax our assumption. All in all, we conclude that the estimate above is robust.

## V. SMEARED REGGE BOUNDS

Next we derive the Regge bound on a smeared amplitude (1.6). Since the smearing goes all the way to $t = 0$, where the amplitude has a pole $\frac{s^2}{t}$, there is a nontrivial interplay between taking $s$ large and smearing over $t$ all the way to $t = 0$.

We will use the following lemma that bounds the smeared $d$-dimensional Legendre polynomials

**Lemma 2:** The averaged $d$-dimensional Legendre polynomials satisfy

$$P_J^{(d)}[\psi_{\mathsf{a},\mathsf{b}}] \leq \min\left( C_1, C_2 \max\left( \left(\frac{J}{\sqrt{s}}\right)^{-2-\mathsf{a}}, \left(\frac{J}{\sqrt{s}}\right)^{\frac{1-d}{2}-\mathsf{b}} \right) \right),$$
(5.1)

where $C_i$ do not depend on $s$ and $J$, and $d \geq 3$. As before we have not proved this lemma, but derived it at large $J$ and checked it numerically. Note that the validity of this Lemma requires the smoothness $C^{\min(\mathsf{a},\mathsf{b})}$ of $\psi_{\mathsf{a},\mathsf{b}}$.[25]

We now proceed as before: we first consider the bound along the $s$-channel cut and split the sum over $J$'s to low spins $J \leq J_*$ and high spins $J > J_*$. Let us first consider the more conservative Born model, where we only use the ansatz for partial waves above $J_{\text{Born}}(s, \epsilon)$. We estimate the low-spin contribution using unitarity

$$|T_{\psi_{\mathsf{a},\mathsf{b}}}^{J \leq J_{\text{Born}}}(s)| \leq \sum_{J=0}^{J_{\text{Born}}} \tilde{n}_J^{(d)} |f_J(s)| \left| P_J^{(d)}[\psi_{\mathsf{a},\mathsf{b}}] \right|$$
$$\leq s^{2-\frac{d}{2}} \sum_{J=0}^{J_{\text{Born}}} \tilde{n}_J^{(d)} \left| P_J^{(d)}[\psi_{\mathsf{a},\mathsf{b}}] \right|$$
$$\lesssim s^{2-\min(1, \frac{\mathsf{a}}{d-4}, \frac{\mathsf{b}+\frac{d-5}{2}}{d-4})},$$
(5.2)

where the $O(s)$ contribution comes spins $J \lesssim \sqrt{s}$; the $\mathsf{a}$, $\mathsf{b}$ dependence arises from $\sqrt{s} \lesssim J \leq J_{\text{Born}}(s, \epsilon)$ and Lemma 2 (5.1). The estimate of the high-spin contribution $J > J_{\text{Born}}(s, \epsilon)$ (following the steps identical to the ones in the previous section) does not change (5.2).[26]

Next we repeat the same computation in the Eikonal+tidal model. For the low-spin contribution we

get in exactly the same manner

$$|T_{\psi_{\mathsf{a},\mathsf{b}}}^{J \leq J_{\text{tidal}}}(s)| \lesssim s^{2-\min(1, \frac{\mathsf{a}+2}{d-2}, \frac{\mathsf{b}+\frac{d-1}{2}}{d-2})}.$$
(5.3)

And we obtain the following estimate from the high-spin contribution $J > J_{\text{tidal}}(s, \epsilon)$

$$|T_{\psi_{\mathsf{a},\mathsf{b}}}^{J > J_{\text{tidal}}}(s)| \lesssim s^{2-\min(\frac{\mathsf{a}}{d-4}, \frac{\mathsf{b}+\frac{d-1}{2}}{d-2})}.$$
(5.4)

Note that the $\mathsf{a}$-dependent piece is the same as in the Born estimate (5.2). This is due to the fact that the dominant contribution for the corresponding sum over spins comes from $J \sim J_{\text{Born}}$. In $d = 5$, the estimates in the Eikonal+GW model are analogous and we do not present them here.

As a side comment, let us remark that in both models we get that the amplitude on the $s$-channel cut satisfies

$$T_{\psi_{\mathsf{a} \geq d-4, \mathsf{b} \geq \frac{d-3}{2}}}(s) \leq C_{\mathsf{a},\mathsf{b}}^{A,B \to A,B} \, s.$$
(5.5)

We now proceed as before and bound the scattering amplitude on the $u$-channel cut. Here we encounter an important subtlety. Using crossing (2.4) we can write the smeared partial wave expansion on the $u$-channel cut[27]

$$T_{\psi_{\mathsf{a},\mathsf{b}}}(-s + i\epsilon) = \sum_{J=0}^{\infty} \tilde{n}_J^{(d)} \int_0^{q_0} dq \, q \psi_{\mathsf{a},\mathsf{b}}(q)$$
$$\tilde{f}_J^*(s + q^2) P_J^{(d)}\left(1 - \frac{2q^2}{s + q^2}\right),$$
(5.6)

where we used different $\tilde{f}_J(s)$ to emphasize that these are the partial wave for the $A, \bar{B} \to A, \bar{B}$ scattering process.

A new feature compared to the $s$-channel cut analysis is that smearing involves both the Legendre polynomials and the partial waves and is therefore not purely kinematical. Thus, we cannot apply (5.1) in the same way without making extra assumptions about the behavior of partial waves $\tilde{f}_J^*(s)$.[28]

We follow the same steps as for the $s$-channel cut. For low spins we get

$$|T_{\psi_{\mathsf{a},\mathsf{b}}}^{J \leq J_*}(-s)| \leq$$
$$\sum_{J=0}^{J_*} \tilde{n}_J^{(d)} \int_0^{q_0} dq \, q |\psi_{\mathsf{a},\mathsf{b}}(q)| |\tilde{f}_J^*(s + q^2)| \left| P_J^{(d)}\left(1 - \frac{2q^2}{s + q^2}\right) \right|$$
$$\leq C s^{2-\frac{d}{2}} \sum_{J=0}^{J_*} \tilde{n}_J^{(d)} \int_0^{q_0} dq \, q^{1+\mathsf{a}} \left| P_J^{(d)}\left(1 - \frac{2q^2}{s + q^2}\right) \right|,$$
(5.7)

---

[25] We thank Massimo Porrati for a discussion on this point.

[26] Using the weakest assumption (4.8) to derive the estimate requires $\mathsf{b} > \frac{d-5}{2}$, and thus $\mathsf{b} > 0$ in $d = 5$. This condition is also needed to establish 2SDR in the Born model (5.15).

[27] The smeared partial wave expansion in the $u$-channel cut can be established in exactly the same way as we did it in the $s$-channel in Appendix C.

[28] Such an extra assumption was implicitly made in the quick derivation of the Regge bound in [17]. The estimate (5.1) was applied there in the presence of an unknown phase $\frac{f_J^*(s+q^2)}{|f_J^*(s+q^2)|}$. Alternatively, we will first establish and then use dispersion relations to derive the desired bounds.

where we used the fact that $|\psi_{\mathsf{a},\mathsf{b}}(q)| \leq Cq^{\mathsf{a}}$. To evaluate the last integral we split it into two parts $\int_0^{\epsilon\frac{\sqrt{s}}{J}} + \int_{\epsilon\frac{\sqrt{s}}{J}}^{q_0}$, and we use the bound (4.5) to estimate the integrals. As a result the integral in (5.7) is bounded as

$$\int_0^{q_0} dq\ q^{1+\mathsf{a}} \left| P_J^{(d)}\left(1 - \frac{2q^2}{s+q^2}\right) \right|$$
$$\leq \min\left(C_3, C_4 \max\left(\left(\frac{J}{\sqrt{s}}\right)^{-2-\mathsf{a}}, \left(\frac{J}{\sqrt{s}}\right)^{\frac{3-d}{2}}\right)\right). \tag{5.8}$$

Using (5.8) we can estimate the low-spin contribution in the various models in the same way as we did before. Finally, for the high-spin part, we can estimate (5.6) using the known behavior of the partial waves. Note that in contrast to the pointwise derivation, the difference between (5.1) and (5.8) induces a different bound along the $s$- and the $u$-channel cuts.

Finally, we should also not forget that in $d = 5$ the leading inelastic effect is due to the emission of gravity waves, and here we use the Eikonal+GW model, where we apply the tree-level eikonal model for $b \geq b_{\mathrm{GW}}(s, \epsilon)$.

The conclusion of this analysis is that we get the following bounds along the real axis, directions (a) and (b) in Figure 2,

$$\mathrm{Born}\big|_{d\geq 5}: \quad |T_{\psi_{\mathsf{a},\mathsf{b}}}(s)| \lesssim |s|^{2-\min(\frac{d-7}{2(d-4)}, \frac{\mathsf{a}}{d-4})}. \tag{5.9}$$

$$\mathrm{Eikonal + tidal}\big|_{d>5}: \quad |T_{\psi_{\mathsf{a},\mathsf{b}}}(s)| \lesssim |s|^{2-\min(\frac{d-3}{2(d-2)}, \frac{\mathsf{a}}{d-4})}. \tag{5.10}$$

$$\mathrm{Eikonal + GW}\big|_{d=5}: \quad |T_{\psi_{\mathsf{a},\mathsf{b}}}(s)| \lesssim |s|^{2-\min(\frac{1}{5}, \mathsf{a})}. \tag{5.11}$$

As for the pointwise case, we can now combine the estimates above with the subexponentiality assumption to conclude that they also apply everywhere in the complex $s$-plane. It follows that the smeared amplitude admits dispersion relations with a few subtractions (2SDR for the Eikonal+tidal/GW model and for the Born model in $d > 7$; 3SDR for the Born model in $d = 6, 7$; 4SDR for the Born model in $d = 5$). A natural question to ask is the following: can we improve the estimates above using dispersion relations?

It is indeed the case due to an improvement of the $u$-channel estimates. Dispersion relations make the smearing in the $u$-channel purely kinematical (exactly as in the $s$-channel). We will provide the details of this mechanism in the next section, but here let us simply quote the final bounds in different models

$$\mathrm{Born + DR}\big|_{d\geq 5}: \ |T_{\psi_{\mathsf{a},\mathsf{b}}}(s)| \lesssim |s|^{2-\min(1, \frac{\mathsf{a}}{d-4}, \frac{\mathsf{b}+\frac{d-5}{2}}{d-4})}. \tag{5.12}$$

$$\mathrm{Eik. + tidal + DR}\big|_{d>5}: \ |T_{\psi_{\mathsf{a},\mathsf{b}}}(s)| \lesssim |s|^{2-\min(1, \frac{\mathsf{a}}{d-4}, \frac{\mathsf{b}+\frac{d-1}{2}}{d-2})}. \tag{5.13}$$

$$\mathrm{Eik. + GW + DR}\big|_{d=5}: \ |T_{\psi_{\mathsf{a},\mathsf{b}}}(s)| \lesssim |s|^{2-\min(1, \mathsf{a}, \frac{2\mathsf{b}+3}{5}\mathsf{b})}. \tag{5.14}$$

Effectively, the improved bounds agree with the ones coming from the estimates in the $s$-channel.

An immediate consequence of the formulas above is that even with the most conservative assumptions of the Born model the smeared amplitude admits dispersion relations with two subtractions

$$\lim_{|s|\to\infty} \frac{T_{\psi_{\mathsf{a},\mathsf{b}}}(s)}{|s|^2} = 0, \quad d \geq 5. \tag{5.15}$$

In $d = 5$, in the Born model, this requires $\mathsf{b} > 0$. In the Eikonal+tidal/GR model, 2SDR hold in a wider range of $\mathsf{b}$'s which always include $\mathsf{b} = 0$.

Another immediate consequence of the formulas above is that in all models

$$\lim_{|s|\to\infty} |T_{\psi_{\mathsf{a}\geq d-4, \mathsf{b}\geq\frac{d-3}{2}}}(s)| \leq C|s|. \tag{5.16}$$

Let us briefly comment on the difference between the formulas above and the pointwise in $t$ bounds derived in the previous section. The bounds in the previous section were derived for fixed $t < 0$. Let us imagine that the scattering amplitude in the Regge limit takes the form

$$\lim_{s\to\infty} T(s,t) \sim g(s,t)s^{j(t)}, \tag{5.17}$$

where $g(s,t)$ is the function of slow variation.[29] The point-wise analysis of the previous section then directly bounds $j(t)$ for $t < 0$.

If we could extrapolate these bounds from $t < 0$ to $t \leq 0$ then the smeared amplitude should satisfy very similar bounds, so why are the smeared bounds different? The point is that this extrapolation does not hold, or, in other words, the limits $t \to 0$ and $s \to \infty$ do not commute. For fixed $s$ the $t \to 0$ limit is controlled by the tree-level result (3.2) which behaves as $s^2$ at large $s$. This nontrivial interplay can be already seen in our estimates of the Legendre polynomials (4.5) which depend on $\frac{s}{q^2}$, where $t = -q^2$. This is the reason why the Regge bounds on the smeared amplitude above depend nontrivially on the properties of the smearing function.

### A. Smeared dispersion relations

At the end of the previous section we stated that dispersion relations can be used to improve the bounds for the smeared amplitude.

We now demonstrate how to go from the bounds (5.9), (5.10), (5.11) to the improved bounds (5.12), (5.13), (5.14). To keep the discussion simple let us

———

[29] The function of slow variation is defined as $\lim_{s\to\infty}\frac{g(\lambda s)}{g(s)} = 1$, see [53]. $(\log s)^k$ is a simple example of this type.

consider the Eikonal+tidal/GW model for the smeared amplitude, where the number of subtractions is always two. Starting with any different number of subtractions does not change the argument.

The argument proceeds as follows. We choose subtractions to write the dispersive representation of the smeared amplitude

$$T_{\psi_{a,b}}(s) = O(s) + \int_{s_0}^{\infty} \frac{ds'}{\pi} \frac{s^2}{(s')^2} \left( \frac{T_{\psi_{a,b}}(s')}{s'-s} - \frac{T_{\psi_{a,b}}(-s')}{s'+s} \right),$$
(5.18)

where $O(s)$ correspond to subtractions (or contribution of the low-energy arcs, as in Figure 2) and for us it is only important that it is linear in $s$ at large $s$. Inside the integral (5.18) we can improve our estimate of the $u$-channel cut made after (5.6). We get for the $u$-channel contribution to the 2SDR

$$\int_{s_0}^{\infty} \frac{ds'}{\pi} \frac{s^2}{(s')^2} \frac{1}{s'+s} \sum_{J=0}^{\infty} \tilde{n}_J^{(d)}$$
$$\int_0^{q_0} dq \, q \psi_{a,b}(q) \tilde{f}_J^*(s'+q^2) P_J^{(d)} \left(1 - \frac{2q^2}{s'+q^2}\right). \quad (5.19)$$

We change the integration variable $s' = \tilde{s}' - q^2$. This separates averaging over transferred momenta and energies at the cost of the $q^2$-dependent lower integration limit $\tilde{s}' \geq s_0 + q^2$. The lower integration limit is, however, immaterial since it is manifest from (5.18) that the contribution to the amplitude of any finite interval in $s'$ grows at most linearly at large $s$. The relevant region to estimate comes from large $s'$.

We therefore need to estimate the following integral for large $s$ and $s'$

$$\int_{s_0+q_0^2}^{\infty} \frac{d\tilde{s}'}{\pi} \sum_{J=0}^{\infty} \tilde{n}_J^{(d)} \tilde{f}_J^*(\tilde{s}')$$
$$\int_0^{q_0} dq \frac{q \psi_{a,b}(q)}{(\tilde{s}'-q^2)^2} \frac{s^2}{\tilde{s}'-q^2+s} P_J^{(d)} \left(1 - \frac{2q^2}{\tilde{s}'}\right). \quad (5.20)$$

Note that now the smearing becomes purely kinematical: it only acts on the $d$-dimensional Legendre polynomials and not partial waves.

The only new ingredient compared to the $s$-channel is presence of the factor $\frac{1}{(\tilde{s}'-q^2)^2(\tilde{s}'-q^2+s)}$. At large $s$ and $s'$ we can expand this integral in $\frac{q^2}{s}$, $\frac{q^2}{s'}$ with higher terms having extra $\frac{1}{s}$, $\frac{1}{s'}$ suppression. Therefore the leading estimate comes from simply setting $q^2 = 0$ in the extra pre-factor which reduces the problem to the one of the $s$-channel

$$\int_{s_0+q_0^2}^{\infty} \frac{d\tilde{s}'}{\pi} \frac{1}{(\tilde{s}')^2} \frac{s^2}{\tilde{s}'+s} (T_{\psi_{a,b}}^{A,\bar{B}\to A,\bar{B}}(\tilde{s}'))^*. \quad (5.21)$$

By applying the $s$-channel estimates first to $T_{\psi_{a,b}}^{A,\bar{B}\to A,\bar{B}}(\tilde{s}')$ and then to the integral above (5.21) we

recover the improved bounds (5.12), (5.13), (5.14) for the discontinuity integrals in (5.18).

As a final step we need to discuss the subtraction terms. In case of the 2SDR they are $O(s)$ and therefore trivially satisfy the improved bound. What if we would have started with the nSDR? In this case the estimate of the discontinuity integrals does not change, but the subtraction terms start with $O(s^{n-1})$. However, we can evaluate (5.18) on the $s$-channel cut where we know that the amplitude is bounded by the improved bounds (as they correspond to the estimate in the $s$-channel). Moreover, we just argued that the integral over the $u$-channel discontinuity also obeys the same bounds as the $s$-channel. Therefore, by evaluating (5.18) on the $s$-channel, we learn that the subtractions violate the bound unless they start as $O(s)$. This completes the argument.

## B. The Pomeranchuk theorem

Let us quickly review a closely related result, namely the Pomeranchuk theorem. We focus on the case $a \geq d-4$, $b \geq \frac{d-3}{2}$ for which all the improved bound (5.12), (5.13), (5.14) become simply $\lesssim |s|$. Imagine now that the amplitude saturates the bound and grows linearly. Namely, we have for the $s$- and $u$-channels

$$T_{\psi_{a,b}}(s) \simeq C_{a,b}^{A,B\to A,B} s, \quad (5.22)$$

$$T_{\psi_{a,b}}^{A,\bar{B}\to A,\bar{B}}(s) \simeq (C_{a,b}^{A\bar{B}\to A\bar{B}})s. \quad (5.23)$$

Consistency of the linear asymptotic behavior and 2SDR requires

$$C_{a,b}^{A,B\to A,B} = -(C_{a,b}^{A\bar{B}\to A\bar{B}})^*, \quad (5.24)$$

which in particular implies that $\text{Im} C_{a,b}^{A,B\to A,B} = \text{Im} C_{a,b}^{A\bar{B}\to A\bar{B}}$. Indeed, imagine that $C_{a,b}^{A,B\to A,B} \neq -(C_{a,b}^{A\bar{B}\to A\bar{B}})^*$, the dispersive integral (5.18) then generates a term $(C_{a,b}^{A,B\to A,B} + (C_{a,b}^{A\bar{B}\to A\bar{B}})^*)s \log s$ which contradicts the asymptotic behavior (5.22), (5.23).

Therefore if the bound (5.5) is saturated, we see that (5.24) implies the smeared version of the Pomeranchuk theorem

$$\lim_{s\to\infty} \frac{\text{Im} \, T_{\psi_{a,b}}^{A,B\to A,B}(s)}{\text{Im} \, T_{\psi_{a,b}}^{A,\bar{B}\to A,\bar{B}}(s)} = 1, \quad (5.25)$$

where $a \geq d-4$, $b \geq \frac{d-3}{2}$. Eq. (5.25) states that smeared cross-sections of, say, proton-proton and proton-antiproton must asymptotically agree.[30]

———

[30] Similar results can also be derived for a more general high-energy asymptotic of the amplitude, see for example the corresponding discussion in [3].

The conclusion of the discussion above is that if the smeared amplitude behaves $C_{\mathsf{a},\mathsf{b}}^{A,B\to A,B}s$ along the real axis, the same constant $C_{\mathsf{a},\mathsf{b}}^{A,B\to A,B}$ controls the linear growth everywhere in the upper half-plane. We will use this fact in what follows.

### C. Black hole formation

Let us next try to compute the constant $C_{\mathsf{a},\mathsf{b}}^{A,B\to A,B}$. This requires some dynamical input. First, to fully decouple non-universal physics we take strict inequalities $\mathsf{a} > d - 4$ and $\mathsf{b} > \frac{d-3}{2}$. With these conditions all spins that scale as $J \sim s^k$ with $k \neq \frac{1}{2}$ cannot produce linear in $s$ contribution to the amplitude. To see it we use unitarity, Lemma 2, and the same type of estimate as in (5.2). The relevant partial waves are thus $J \sim \sqrt{s}$, which also correspond to scattering at fixed impact parameters $b$ via the relation (3.15).

In a gravitational theory, for any fixed impact parameter $b$, in the limit of large energies we have $b \ll b_{\mathrm{Sch}}(s) \sim s^{\frac{1}{2(d-3)}}$, and the relevant physical picture is that the incoming particles form a black hole with a unit probability.[31] At the level of the exclusive $2 \to 2$ amplitude we thus expect to have large inelastic effects. A maximally inelastic $S$-matrix is described by $S_J(s) \simeq 0$ which corresponds to[32]

$$f_{J\sim b\sqrt{s}}^{\mathrm{BH}}(s) \simeq \frac{\sqrt{s}}{(s - 4m^2)^{\frac{d-3}{2}}}i, \quad b \lesssim b_{\mathrm{Sch}}(s), \quad (5.26)$$

where the corrections to this formula are suppressed at large $s$.[33]

Combining (5.26) with the fact that partial waves in the regime outside of its validity necessarily produce a sub-linear contribution at large $s$, we arrive at the following formula for the asymptotic constant $C_{\mathsf{a},\mathsf{b}}^{A,B\to A,B}$

$$C_{\mathsf{a},\mathsf{b}}^{A,B\to A,B} = \lim_{s\to\infty} \frac{1}{s} \sum_{J=0}^{\infty} \tilde{n}_J^{(d)} f_J^{\mathrm{BH}}(s) P_J^{(d)}[\psi_{\mathsf{a},\mathsf{b}}]$$

$$\propto \int_0^{q_0} dq\, q^{4-d}\psi_{\mathsf{a},\mathsf{b}}(q)\delta(q) = 0, \quad \mathsf{a} > d - 4, \quad (5.27)$$

---

[31] Together with an $O(1)$ fraction of incoming energy that goes into radiation [54].

[32] Such a behavior is also known as a black disk model, and it is possible that it is realized in QCD with $R_{\mathrm{disk}}(s) \sim \log s$ [20–22] (it is realized in holographic QCD [18, 19]). As long as the black disc model is a correct description of the large energy asymptotic of the fixed impact parameter scattering our conclusions that follow will hold. In other words, the fact that it is due to production of black holes is not essential for the argument.

[33] A common expectation is that in this regime schematically we have for the phase shift $\mathrm{Im}\delta_{J\sim b\sqrt{s}}(s) \sim S_{BH}(\sqrt{s})$, see [29, 55–57]. But we will not need the precise suppression factor for our purposes.

where in the second line we used the completeness relation for the $d$-dimensional Legendre polynomials (A.4).

The conclusion of this discussion is that the gravitational amplitudes in fact admit 1SDR, namely we have

$$\text{Born+BH}: \quad \lim_{|s|\to\infty} \frac{T_{\psi_{\mathsf{a}>d-4,\mathsf{b}>\frac{d-3}{2}}}(s)}{|s|} = 0. \quad (5.28)$$

The conditions $\mathsf{a} > d - 4$, $\mathsf{b} > \frac{d-3}{2}$ guarantee the suppression of the non-universal contributions making them sub-linear, and the fixed impact parameter contribution is sub-linear due to (5.26) and (5.27). It would be interesting to explore the consequences of dispersive sum rules that follow from (5.28). It would be also interesting to study analogous sum rules in AdS/CFT.

## VI. LOCAL GROWTH BOUNDS

In the sections above, we discussed the asymptotic behavior of gravitational scattering amplitudes. Since physical experiments are always performed at finite energies, it is very interesting to discuss the bound on the *local* growth of the amplitude. Existence of such a bound is expected based on very general grounds of causality and unitarity [4].[34] Here we derive bounds on the local growth of the scattering amplitudes starting from dispersion relations. A toy model for establishing such a bound was discussed in [60] and we follow the same basic idea.

Let us first consider scattering of identical particles in gapped theories.[35] In this case, it is easy to show, see Appendix D, that

$$-3 \leq y\partial_y \log \mathrm{Im}\Big[T(s_0 + iy, t)\Big] \leq 1, \quad 0 \leq t < m_{\mathrm{gap}}^2,$$
$$(6.1)$$

where $m_{\mathrm{gap}}^2$ is the mass-square of the lightest exchanged state in the $t$-channel, or, equivalently, the location of the first nonanalyticity for $t > 0$. The constraint on $t$ in the bound above guarantees positivity of the discontinuity of the amplitude that enters into the relevant dispersion relations, see Appendix D.

To understand the implications of (6.1), let us model the local behavior of the amplitude via a pair of complex conjugate Regge poles $T(s,t) \sim f(t)[(-is)^{j(t)} + $

---

[34] In the context of AdS/CFT the corresponding bound is related to the bound on chaos, see appendix A in [23] (see also [58, 59]). The relation of the bound on chaos to the local growth of the flat space amplitudes was recently explored in [24].

[35] We thank Amit Sever for discussions on this topic.

$(-is)^{j^*(t)}$.[36] Reality property of the amplitude along the imaginary axis (which is a consequence of unitarity and crossing) requires that $f(t) \in \mathbb{R}$.

Let us consider separately the cases when $j(t)$ is complex and when it is real. In the complex case, $\text{Im}[j(t)] \neq 0$, (6.1) leads to the bound

$$|\text{Re}[j(t)]| \leq 1, \quad \text{Im}[j(t)] \neq 0, \quad 0 \leq t < m_{\text{gap}}^2. \quad (6.2)$$

For the case when $\text{Im}[j(t)] = 0$, (6.1) leads to the bound

$$|j(t)| \leq 2, \quad \text{Im}[j(t)] = 0, \quad 0 \leq t < m_{\text{gap}}^2. \quad (6.3)$$

We can apply (6.3) for example to the large $N$ QCD, where the local behavior of the amplitude is controlled by the leading planar amplitude $T_{\text{planar}}(s, t)$. The bound (6.3) then constrains the planar Regge trajectory and states that the planar amplitude cannot grow faster than $s^2$ for $0 \leq t < m_{\text{gap}}^2$. The bound (6.3) is also trivially consistent with the Donnachie-Landshoff pomeron fit $j(0) \approx 1.08$ of the hadronic total cross sections [61]. Similarly, (6.3) can be used to test the Regge models for elastic scattering of hadrons away from $t \neq 0$, see e.g. [62].

Next, we would like to generalize the discussion above to $t \leq 0$ and gravitational theories. We will restrict our attention only to the weakly coupled theories.[37] By assumption in these theories the scattering amplitude admits an expansion in the small coupling $g \ll 1$ and we will be interested in the leading behavior of the amplitude $O(g)$. This small parameter can be the Planck mass for gravitational theories, or the large number of colours in QCD. In this context it is also natural to introduce another notion of $M_{\text{gap}}^2$ which describes the gap not in the nonperturbative amplitude, but in the leading $O(g)$ amplitude, see e.g. [15]. In the context of weakly coupled string theory $M_{\text{gap}}^2$ describes the string scale, whereas in the large $N$ QCD it is related to $\Lambda_{QCD}$ and describes the mass of the lightest glueball exchanged in the planar amplitude.

Deriving the bound on the local growth of the amplitude for fixed $t < 0$ using dispersion relations is problematic, because the discontinuity of the amplitude is not nonnegative in this case. For this reason we focus on the smeared amplitude and look for functionals which effectively make the smeared discontinuity of the amplitude nonnegative. It is precisely the same strategy, as the one pursued in the context of bounding the Wilson coefficients in [15].

More precisely, we consider dispersion relations with two subtractions, as in (5.15). We fix $s = s_0 + iy$ and

search for the smearing function $\psi(q)$ such that

$$-3 \leq y\partial_y \log \text{Im}\left[T_\psi(s_0 + iy)\right] \leq 1. \quad (6.4)$$

Indeed, given $s_0$ and $y$ we were able to find the smearing function $\psi(q)$ such that (6.4) holds. We present the details of our search in Appendix E. In this search, we set $M_{\text{gap}} = 1$ and choose to smear up to $q_0 = 1$. As a result, we found the following functionals

$$d = 4|_{\text{short-range}} : \quad \psi(q) = \frac{(1-q)^2}{q}, \quad (6.5)$$

$$d \geq 5 \quad : \quad \psi(q) = q(1-q^2)(1-q)^2, \quad (6.6)$$

Where $|_{\text{short-range}}$ in (6.5) emphasizes that the functional can only be applied to theories without long-range forces (i.e. no $1/t$ pole). In $d = 4$, we have also found the functional $\psi(q) = (1 - q)^2$. This functional covers a smaller region than (6.5) in the $s$-complex plane. However, if we consider a gravitational theory, this functional will produce a logarithmic divergence due to the graviton pole (in contrast to a power-law divergence coming from (6.5)). This logarithmic divergence can be presumably regulated by considering an IR-safe observable, see e.g. [16, 17].

For each functional in (6.5), (6.6) we present in Figure 5 the region in the complex $s$-plane for which (6.4) holds. These functionals cover a large region of the complex $s$-plane, but may not be optimal in the sense that we could find a functional for which (6.4) holds in a larger region.

The local bound (6.4) is interesting because it constrains the Regge limit of the tree-level amplitudes, e.g. in the tree-level string theory or the large $N$ QCD.[38] Indeed, since the theory is weakly coupled in the large range of energies the full amplitude is well approximated by the tree-level result $T_{\text{tree}}(s, t)$. The bound (6.4) then controls the Regge limit of $T_{\text{tree}}(s, t)$.

To better understand its implication, let us imagine again that the amplitude locally takes the form $T_{\text{tree}}(s, t) \sim f(t)(-is)^{j(t)}$, $f(t) \in \mathbb{R}$. Considering $y \gg s_0$ and taking $j(t) \in \mathbb{R}$, the bound (6.4) becomes

$$|\langle j(t) \rangle_\psi| = \left| \frac{\int_{-M_{\text{gap}}^2}^0 dt \, \psi(t) j(t) f(t) y^{j(t)-1}}{\int_{-M_{\text{gap}}^2}^0 dt \, \psi(t) f(t) y^{j(t)-1}} \right| \leq 2, \quad (6.7)$$

where we used that $\text{Im}(y - is_0)^{j(t)} \simeq -y^{j(t)-1} s_0$.

Defining $j_{\max} \equiv \max_{-M_{\text{gap}}^2 \leq t \leq 0} j(t)$, we see that (6.7) implies that for large $y$, this integral is dominated by the region close to $j_{\max}$ and the constraint becomes[39]

$$|j_{\max}| \leq 2, \quad (6.8)$$

---

[36] Considering Regge poles versus more complicated singularities is not essential for the argument. In particular, adding powers of $\log s$ in the ansatz for the amplitude does not change any of the conclusions.

[37] This is not strictly necessary as we briefly discuss in the conclusions.

---

[38] We explicitly check that (6.4) holds for the Virasoro-Shapiro amplitude in Appendix E 2.

[39] Here we assumed no cancelations so that the integral is dominated by $j(t) = j_{\max}$.

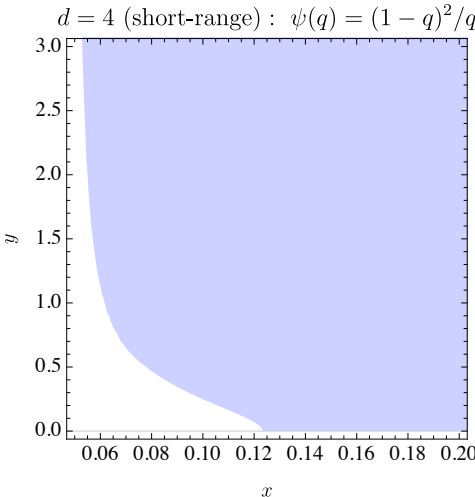
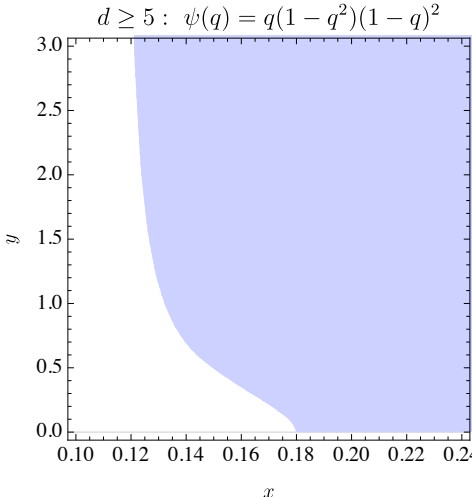

FIG. 5. Regions of validity of the bound (6.4) for the functionals (6.5) and (6.6) in the $s$-complex plane, where $s_0 = 2m^2 + x$. For the detailed plots and explanations of the underlying procedure, see Appendix E. The units are fixed by choosing $M_{\rm gap} = 1$ and we chose to smear up to $q_0 = 1$.

which is precisely the statement of the CRG conjecture [25] applied to scalar particles and $-M_{\rm gap}^2 \leq t \leq 0$. We also see that for the gapped theories the CRG conjecture naturally includes the region of positive $t$ as in (6.3).

Finally, let us discuss the assumption at the beginning of this section that the scattered particles are identical. This was needed in the derivation of the local growth bound since we used the $s-u$ channel crossing symmetry. A simple generalization to the case of nonidentical particles is to consider

$$T(s,t) \equiv T^{A,B \to A,B}(s,t) + T^{A,\bar{B} \to A,\bar{B}}(s,t), \qquad (6.9)$$

for which the same argument automatically applies. It would be interesting to generalize the argument of this section to more general situations.

Note also that every set of $s - u$ symmetric degrees of freedom that separately satisfies 2SDR generates a scattering amplitude that can only produce time delay. This statement is in the spirit of the recent discussion of infrared causality in [63, 64].

## VII. CONCLUSIONS AND FUTURE DIRECTIONS

In this paper we analyzed the Regge behavior of the $2 \to 2$ elastic scattering amplitude in a relativistic, gravitational theory. We reviewed the standard assumptions about the properties of the nonperturbative amplitude and used them to derive bounds on the Regge limit. Our results are summarized in Table I and we expect that the pointwise bounds in the Eikonal+tidal model are saturated. Our basic conclusion is that the scattering amplitude (both pointwise and smeared over the transferred momentum) admits twice-subtracted dispersion relations (2SDR).

We discussed a few additional observations starting from 2SDR:

- We used the classical picture of black hole formation at high energies and fixed impact parameters to argue that the number of subtractions for a sub-class of smeared amplitudes is one (1.10). Our argument could be applicable in the non-gravitational context as well.

- We derived a bound on the local growth of the amplitude as a function of energy. When applied to weakly coupled gravitational theories and large $N$ gauge theories, our result establishes a scalar version of the CRG conjecture [25], and generalizes it to the gapped theories.

Our results agree with the analysis of [16], where the rigorous AdS bounds were derived and they reduced to the bounds derived from 2SDR in the flat space limit. This result supports nonperturbative validity of our assumptions.

A bound on the Regge behavior of gravitational scattering amplitudes is a starting point for studying dispersion relations, which has been the subject of many recent works [65–91]. We hope that our analysis could be useful for further explorations of the gravitational dispersion relations and their consequences.

Let us next comment on some of the obvious open questions and further directions.

### Legendre estimates

An important ingredient of our work are the estimates of the $d$-dimensional Legendre polynomials at large spin. These are encapsulated by two lemmas, (4.5) and (5.1). We used them extensively, but we have not proved them. It would be very useful to prove them and moreover

generalize them to the spinning Legendre polynomials, see e.g. [73, 92]. It could be also interesting to develop similar techniques in AdS.

### Spinning particles

An obvious, but exciting extension of the present analysis is to consider scattering of spinning particles, e.g. photons and gravitons [17, 73, 77, 84, 92–94]. The main ingredients in deriving the spinning bounds will stay the same, in particular our working assumptions will remain unchanged. The major difference lies in the presence of multiple polarizations and the corresponding need to consider a matrix of scattering amplitudes. This would allow us to explore bounds on the local growth of the spinning amplitudes. It would require a generalization of the analysis in section VI of the present paper. Derivation of such a bound on the local growth of the amplitude would constitute a dispersive derivation of the CRG conjecture for spinning particles.

### Dispersion relations with one subtraction

We have observed that in gravitational theories there exists a sub-class of smearing functions $\psi_{a,b}(q)$ for which the scattering amplitude admits 1SDR, see (1.10). It would be interesting to explore the consequences of new sum rules that arise from this fact. Given that the number of subtractions in QFT on general grounds is two, it means that when we couple QFT to gravity the set of low-energy couplings that admit dispersive representation becomes larger. A similar phenomenon takes place when we increase the spin of external particles and is known as superconvergence [95]. It would be fascinating to explore the dispersive consequences of coupling the Standard Model to gravity in more detail.[40] It would be also interesting to explore 1SDR in the context of AdS/CFT, where we expect that our argument still works based on the correspondence between the flat space and AdS dispersive sum rules established in [16].

### Local growth bound

The local bound on the growth of the amplitude derived in this paper, namely that the amplitude does not grow faster than $s^2$, appears in many different contexts. In flat space, it appeared in the scattering experiment considered in [4] and in the closely related CRG conjecture [24, 25]. In AdS, the same bound on the Regge behavior of the correlation function in the planar theory exists [23]. For the thermal correlator, the same power $s^2$ controls scattering close to the black hole horizon [99], and the local growth bound becomes the bound on the Lyapunov exponent in the dual CFT [23]. A more general result along the same lines is known as

the modular chaos bound [100]. Assuming that the local growth bound holds generally, it would be interesting to understand what are its full implications and what are the EFT constraints that can be extracted from it.

Coming back to our analysis, we have found simple functionals (6.5) and (6.6) that provide a local bound on scattering in a large region of the complex $s$-plane. For example, in Appendix E 2 we applied the local growth bound to the Virasoro-Shapiro amplitude and observed that it satisfies the bound in a larger portion of the $s$-plane. It would be interesting to see if there exists a functional for which it lives closer to the boundary of the allowed region. Similarly, one could search for a functional valid in the largest possible region in the $s$-plane.

In the discussion of the smeared local growth bound, we chose to work with the assumption that the theory is weakly coupled. However, this is not strictly necessary. The argument for this is the same as in Section V A, namely that any finite integration region in the dispersion integral can only produce terms $O(s)$, and as such cannot affect the $\lesssim s^2$ bound. Therefore, even if the integral over the discontinuity in the dispersion relations starts at $s_0$, as opposed to $M_{gap}^2$, we can absorb the contribution of the part of the cut $(s_0, M_{gap}^2)$ into the definition of the amplitude and run our argument. Since the subtraction terms in 2SDR are $O(s)$ it will not affect the bound $j_{max} \leq 2$.

### Four dimensions

A very interesting problem is to generalize our analysis to $d = 4$. An immediate problem is that $T(s,t)$ is not a well-defined observable in four dimensions. Relatedly, the nonperturbative unitarity condition (3.13) needs to be formulated for the corresponding IR finite observables (possibly along the lines of [26–29]). Finally, all the other assumptions such as analyticity, subexponentiality and crossing will have to be re-evaluated.

Nevertheless, since the origin of difficulties in $d = 4$ lies in the large distance physics which is, in principle, under control,[41] we believe that the basic idea used in this note, namely to use nonperturbative unitarity and known large impact parameter physics to constrain the Regge limit, should eventually work. More concretely, let us assume that we can define an IR-regulated smeared amplitude $T_{\psi_{a,b}}^{IR}(s)$ and the corresponding partial waves $f_J^{IR}(s)$. They can explicitly depend on the IR-regulator which affects partial waves especially at large $J$ (related to the large-distance physics), however let us assume that $f_J^{IR}(s)$ satisfy the same nonperturbative unitarity condition as we used in the paper. It is then possible to estimate the behavior of this IR-regulated

-------

[40] Dispersion relations with one subtraction have been explored in the context of electroweak physics for instance in [96–98].

-------

[41] In other words, gravitational clustering as we defined it still holds. However, since $b_{Born} = \infty$ in $d = 4$ the usual clustering breaks down.

observable using the Eikonal+GW model. As a result, it admits 3SDR. We can use dispersion relations with three subtractions to improve the $u$-channel estimate and conclude that $|T^{\mathrm{IR}}_{\psi_{\mathsf{a}>0,\mathsf{b}\geq\frac{1}{2}}}(s)| \lesssim s$ so the number of subtractions in this case is in fact two as well. Finally, the black hole production argument goes through, and we conclude that $T^{\mathrm{IR}}_{\psi_{\mathsf{a}>0,\mathsf{b}>\frac{1}{2}}}(s)$ admits 1SDR.

AdS/CFT provides a natural IR regulator to the four-dimensional flat space scattering problem. A detailed analysis of the flat space limit of the corresponding CFT correlators is consistent with the qualitative conclusions of the previous paragraph [16, 17].[42]

Yet another possible approach is to explore perturbative amplitudes in $d = 4$ focusing on the interesting IR finite contributions (for example the tree-level string amplitudes or the one-loop integrated matter). For these IR finite perturbative amplitudes the notions of crossing, analyticity, and perturbative unitarity are well-understood, and therefore one can study dispersion relations and derive various bounds,

see e.g. [73, 84, 93, 94]. These are expected to control the leading corrections to the properly defined nonperturbative observables.

*Acknowledgments*

We thank Nima Arkani-Hamed, Brando Bellazzini, Simon Caron-Huot, Giulia Isabella, Gregory Korchemsky, Shiraz Minwalla, Pier Monni, João Penedones, Leonardo Rastelli, Riccardo Rattazzi, Francesco Riva, Slava Rychkov, Amit Sever, David Simmons-Duffin, Piotr Tourkine, and Gabriele Veneziano for useful discussions. We thank Gabriele Veneziano for comments on the manuscript. This project has received funding from the European Research Council (ERC) under the European Union's Horizon 2020 research and innovation programme (grant agreement number 949077).

## Appendix A: Legendre polynomials

We follow the conventions of [47]. The $d$-dimensional Legendre polynomials are defined by

$$P_J^{(d)}(z) \equiv {}_2F_1(-J, J + d - 3, \frac{d-2}{2}, \frac{1-z}{2}) = \frac{\Gamma(1+J)\Gamma(d-3)}{\Gamma(J+d-3)} C_J^{(\frac{d-3}{2})}(z), \tag{A.1}$$

where $C_J^{(\frac{d-3}{2})}(z)$ are the standard Gegenbauer polynomials.

With this definition we have

$$|P_J^{(d)}(z)| \leq 1, \quad |z| \leq 1. \tag{A.2}$$

These polynomials satisfy the orthogonality relation

$$\frac{1}{2}\int_{-1}^{1} dz \, (1-z^2)^{\frac{d-4}{2}} P_J^{(d)}(z) P_{\tilde{J}}^{(d)}(z) = \frac{\delta_{J\tilde{J}}}{\mathcal{N}_d \, n_J^{(d)}} , \tag{A.3}$$

and the completeness relation

$$\sum_{J=0}^{\infty} n_J^{(d)} P_J^{(d)}(y) P_J^{(d)}(z) = \frac{2}{\mathcal{N}_d}(1-z^2)^{\frac{4-d}{2}}\delta(y-z) , \tag{A.4}$$

with

$$\mathcal{N}_d = \frac{(16\pi)^{\frac{2-d}{2}}}{\Gamma(\frac{d-2}{2})} , \quad n_J^{(d)} = \frac{(4\pi)^{\frac{d}{2}}(d+2J-3)\Gamma(d+J-3)}{\pi\,\Gamma(\frac{d-2}{2})\Gamma(J+1)} . \tag{A.5}$$

At large $J$ we have $n_J^{(d)} \sim J^{d-3}$.

---

[42] Relatedly, one can substitute $R_{AdS} \to R_{dS}$ in the $AdS_4$ bounds, turning them into quasi-bounds in the terminology of [101], and apply them to our Universe [17]. Eventually we would like to understand the implications of consistency, e.g. no time machines, in all thought experiments performed in a finite region of space and time (and thus finite energy in a gravitational theory), see e.g. [102, 103]. It would be interesting to see the relation of bounds derived in this way and the dispersive bounds. It would be also very interesting to derive dispersive bounds directly in dS, see e.g. [104–106].

## Appendix B: Pointwise convergence of the partial wave expansion

Let us consider a continuous function $f(x)$ that satisfies

$$\int_{-1}^{1} dx (1-x^2)^{\frac{d-5}{4}} |f(x)| < \infty, \quad d \geq 3. \tag{B.1}$$

Then the partial wave expansion converges inside the unit interval

$$f(x) = \sum_{J=0}^{\infty} n_J^{(d)} f_J P_J^{(d)}(x), \quad -1 < x < 1. \tag{B.2}$$

Convergence at the boundary points $x = \pm 1$ requires further conditions on $f(x)$. The condition that $f(x)$ is continuous can be substituted by a weaker condition, see theorem 9.1.2 in G. Szegö.

In the context of scattering amplitudes we are interested in the functions that have a pole close to $x = 1$, so that $|f(x)| \sim \frac{1}{|1-x|}$. The convergence criterion (B.1) is satisfied for $d > 5$, therefore we conclude that the partial wave expansion of the gravitational scattering amplitude converges pointwise for $d > 5$. In $d = 5$, they do not converge pointwise but they do as a distribution, see e.g. [46] for a related discussion.

## Appendix C: Smeared partial wave expansion

In this appendix we would like to establish (3.9). We will first discuss the case when the amplitude admits a convergent partial wave expansion, namely $d > 5$. We then generalize the discussion to $d \geq 5$.

We start with a convergent partial wave expansion in $d > 5$

$$T(s, -q^2) = \sum_{J=0}^{\infty} \tilde{n}_J^{(d)} f_J(s) P_J^{(d)}\left(1 - \frac{2q^2}{s - 4m^2}\right). \tag{C.1}$$

We then consider $\psi(q)$ that satisfies (3.8), and we argue that

$$\int_0^{q_0} dq\, q\, \psi(q) T(s, -q^2) = \sum_{J=0}^{\infty} \tilde{n}_J^{(d)} f_J(s) P_J^{(d)}[\psi], \tag{C.2}$$

where recall that $P_J^{(d)}[\psi] = \int_0^{q_0} dq\, q \psi(q) P_J^{(d)}(1 - \frac{2q^2}{s-4m^2})$, see (3.10). Following [48] we call this the swapping property.

**The swapping property**:

Here we would like to show that

$$T_\psi(s) \equiv \int_0^{q_0} dq\, q [\psi(q) T(s, -q^2)] \overset{?}{=} \sum_{J=0}^{\infty} \tilde{n}_J^{(d)} f_J(s) P_J^{(d)}[\psi]. \tag{C.3}$$

We choose $\epsilon > 0$ and split the integral as follows

$$T_\psi(s) \equiv \int_0^{q_0} dq\, q [\psi(q) T(s, -q^2)] = \int_\epsilon^{q_0} dq\, q [\psi(q) T(s, -q^2)] + \int_0^\epsilon dq\, q [\psi(q) T(s, -q^2)] \tag{C.4}$$

$$= \sum_{J=0}^{\infty} \tilde{n}_J^{(d)} f_J(s) \int_\epsilon^{q_0} dq\, q \left[\psi(q) P_J^{(d)}\left(1 - \frac{2q^2}{s - 4m^2}\right)\right] + \int_0^\epsilon dq\, q [\psi(q) T(s, -q^2)],$$

where in the second line we used that away from the forward limit $q = 0$, the expansion in Legendre polynomial converges to $T(s, -q^2)$, as in (B.2), which is by assumption uniformly bounded for $\epsilon \leq q \leq q_0$. Therefore we can use the dominated convergence theorem to justify the swapping. Alternatively in $d > 7$ we can use the Fubini-Tonelli theorem since the partial wave expansion converges absolutely. Finally, we can avoid using any theorems by explicitly splitting the sum into low spins ($J \leq J_*$) and high spins ($J > J_*$); swapping the sum and the integral in a finite low spin sum; showing that the the high spin tail goes to zero as $J_* \to \infty$. We follow this latter path below.

Regarding the second term in the second line of (C.4) we notice that

$$\lim_{\epsilon \to 0} \int_0^\epsilon dq \, q[\psi(q)T(s,-q^2)] = 0, \tag{C.5}$$

as long as $\psi(q)$ satisfies (3.8). To argue for (C.5) at fixed $s$ we can choose $\epsilon$ to be small enough so that the amplitude is controlled by the graviton pole (3.2). The integral is then $O(\epsilon^{\mathsf{a}})$ and goes to zero in the limit $\epsilon \to 0$ since $\mathsf{a} > 0$.

In this way we end up with the following expression

$$T_\psi(s) = \lim_{\epsilon \to 0} \sum_{J=0}^\infty \tilde{n}_J^{(d)} f_J(s) \int_\epsilon^{q_0} dq \, q \left[ \psi(q) P_J^{(d)} \left( 1 - \frac{2q^2}{s - 4m^2} \right) \right]. \tag{C.6}$$

Next we would like to swap $\lim_{\epsilon \to 0}$ and $\sum_{J=0}$. To do so, let us split the sum over spins:

$$\lim_{\epsilon \to 0} \sum_{J=0}^\infty \tilde{n}_J^{(d)} f_J(s) \int_\epsilon^{q_0} dq \, q \left[ \psi(q) P_J^{(d)} \left( 1 - \frac{2q^2}{s - 4m^2} \right) \right]$$

$$= \sum_{J=0}^{J_*} \tilde{n}_J^{(d)} f_J(s) P_J^{(d)}[\psi] + \lim_{\epsilon \to 0} \sum_{J=J_*+1}^\infty \tilde{n}_J^{(d)} f_J(s) \int_\epsilon^{q_0} dq \, q \left[ \psi(q) P_J^{(d)} \left( 1 - \frac{2q^2}{s - 4m^2} \right) \right], \tag{C.7}$$

where we commuted the limit $\epsilon \to 0$ with the partial sum. In order to proceed, we consider $J_* \gg 1$ fixed and estimate the contribution of the second term.

Due to (3.6) we have

$$\tilde{n}_J^{(d)} f_J(s) \lesssim J, \quad J \to \infty. \tag{C.8}$$

Recall that it follows from the fact that the large $J$ limit is controlled by the behavior of $T(s,-q^2) \sim q^{-2}$ close to $q = 0$. Therefore we can choose $J_*$ and $C$ such that

$$\tilde{n}_J^{(d)} f_J(s) \leq CJ, \quad J > J_*, \tag{C.9}$$

where $C$ is some constant (it can and does depend on $s$).

Next we bound the integral of the Legendre polynomial. We get the following result

$$\int_\epsilon^{q_0} dq \, q \left[ \psi(q) P_J^{(d)} \left( 1 - \frac{2q^2}{s - 4m^2} \right) \right] \lesssim \max\left( J^{\frac{1-d}{2}}, J^{-2-\mathsf{a}} \right), \tag{C.10}$$

where crucially the bound is uniform in $\epsilon \geq 0$. The bound (C.10) requires some explanation. The integral in the LHS can be explicitly taken and its large $J$ behavior for various $\epsilon$ can be analyzed. In particular, there are several characteristic regimes $\epsilon \sim J^{-\alpha}$ with the final estimate given by (C.10).

Putting together (C.9) and (C.10) we get

$$\left| \sum_{J=J_*+1}^\infty \tilde{n}_J^{(d)} f_J(s) \int_\epsilon^{q_0} dq \, q \left[ \psi(q) P_J^{(d)} \left( 1 - \frac{2q^2}{s - 4m^2} \right) \right] \right| \lesssim \sum_{J=J_*+1}^\infty J \max\left( J^{\frac{1-d}{2}}, J^{-2-\mathsf{a}} \right) \sim \max\left( J_*^{\frac{5-d}{2}}, J_*^{-\mathsf{a}} \right). \tag{C.11}$$

Since the estimate does not depend on $\epsilon$, the limit $\epsilon \to 0$ can be now taken. In this way we get

$$T_\psi(s) = \sum_{J=0}^{J_*} \tilde{n}_J^{(d)} f_J(s) P_J^{(d)}[\psi] + O\left( \max\left( J_*^{\frac{5-d}{2}}, J_*^{-\mathsf{a}} \right) \right). \tag{C.12}$$

Finally, we take the limit $J_* \to \infty$ to get the desired result. This concludes the proof of the swapping property (C.3). It was crucial in the derivation that $\mathsf{a} > 0$ and $d > 5$.

**Generalization to $d = 5$:**

To generalize the argument above to $d = 5$ we consider a regularized amplitude

$$T_{\hat{a}}(s,-q^2) \equiv q^{\hat{a}} T(s,-q^2), \tag{C.13}$$

where $\hat{a} > 0$. This makes the partial wave expansion convergent in $d \geq 5$ and we can write

$$T_{\hat{a}}(s, -q^2) = \sum_{J=0}^{\infty} \tilde{n}_J^{(d)} f_{J,\hat{a}}(s) P_J^{(d)} \left(1 - \frac{2q^2}{s - 4m^2}\right). \tag{C.14}$$

We can then run the argument of the previous section with the only difference being that

$$\tilde{n}_J^{(d)} f_{J,\hat{a}}(s) \lesssim J^{1-\hat{a}}, \quad J \to \infty. \tag{C.15}$$

and where we define the regularized smeared amplitude as

$$T_{\psi,\hat{a}}(s) \equiv \int_0^{q_0} dq \, q[\psi(q) T_{\hat{a}}(s, -q^2)] \tag{C.16}$$

We proceed as before and write similarly to (C.6)

$$T_\psi(s) = \lim_{\hat{a} \to 0} \int_0^{q_0} dq \, q[\psi(q) T_{\hat{a}}(s, -q^2)] = \lim_{\hat{a} \to 0} \lim_{\epsilon \to 0} \sum_{J=0}^{\infty} \tilde{n}_J^{(d)} f_{J,\hat{a}}(s) \int_\epsilon^{q_0} dq \, q \left[\psi(q) P_J^{(d)} \left(1 - \frac{2q^2}{s - 4m^2}\right)\right]. \tag{C.17}$$

Then, we split the sum over $J$'s to low spins and high spins

$$\lim_{\hat{a} \to 0} \lim_{\epsilon \to 0} \sum_{J=0}^{\infty} \tilde{n}_J^{(d)} f_{J,\hat{a}}(s) \int_\epsilon^{q_0} dq \, q \left[\psi(q) P_J^{(d)} \left(1 - \frac{2q^2}{s - 4m^2}\right)\right]$$
$$= \sum_{J=0}^{J_*} \tilde{n}_J^{(d)} f_J(s) P_J^{(d)}[\psi] + \lim_{\hat{a} \to 0} \lim_{\epsilon \to 0} \sum_{J=J_*+1}^{\infty} \tilde{n}_J^{(d)} f_{J,\hat{a}}(s) \int_\epsilon^{q_0} dq \, q \left[\psi(q) P_J^{(d)} \left(1 - \frac{2q^2}{s - 4m^2}\right)\right] \tag{C.18}$$

In $d = 5$, the estimate (C.10) is not sufficient to prove what we want. However it can be refined by the following formula

$$\int_\epsilon^{q_0} dq \, q \left[\psi(q) P_J^{(d=5)} \left(1 - \frac{2q^2}{s - 4m^2}\right)\right] \overset{d=5}{\lesssim} \max\left(\cos(\alpha J) J^{-2}, \epsilon^{\mathsf{a}} \cos(\beta \epsilon J) J^{-2}, J^{-2-\mathsf{a}}\right), \tag{C.19}$$

where $\alpha, \beta$ do not depend on $J$ and $\epsilon$. We then have the following estimate

$$\left| \sum_{J=J_*+1}^{\infty} \tilde{n}_J^{(d=5)} f_{J,\hat{a}}(s) \int_\epsilon^{q_0} dq \, q \left[\psi(q) P_J^{(d=5)} \left(1 - \frac{2q^2}{s - 4m^2}\right)\right] \right| \lesssim \sum_{J=J_*+1}^{\infty} \max\left(\cos(\alpha J) J^{-1-\hat{a}}, \epsilon^{\mathsf{a}} \cos(\beta \epsilon J) J^{-1-\hat{a}}, J^{-1-\hat{a}-\mathsf{a}}\right)$$
$$\lesssim J_*^{-1-\hat{a}} + J_*^{-\mathsf{a}-\hat{a}} \tag{C.20}$$

where to obtain the last line the characteristic regimes $\epsilon \sim J^{-\alpha}$ have to be considered separately.

The limits $\epsilon \to 0$ and $\hat{a} \to 0$ can now be taken and we get:

$$T_\psi(s) = \sum_{J=0}^{J_*} \tilde{n}_J^{(d)} f_J(s) P_J^{(d=5)}[\psi] + O(\max(J_*^{-1}, J_*^{-\mathsf{a}})). \tag{C.21}$$

Finally we take the limit $J_* \to \infty$ to conclude the proof in $d = 5$.

## Appendix D: Local growth bound (gapped case)

In this section we would like to put a bound on the local growth of the scattering amplitude in a gapped theory. The limit on the asymptotic growth of the amplitude is given by (1.1) and by the famous Froissart-Martin bound for $t = 0$. In the spirit of the signal model of [4] and [23] we can ask: *what is the limit on the local growth of the amplitude?* Here we derive such a bound from dispersion relations along the lines of the toy model described in [60]. We assume that the scattering amplitude admits 2SDR for $0 \leq t < m_{\text{gap}}^2$, where $m_{\text{gap}}^2$ is the location of the first nonanalyticity in the $t$-channel. In the simplest case of scattering the lightest particles and no bound states present in the $t$-channel $m_{\text{gap}}^2 = 4m^2$.

We consider a gapped theory and we consider 2SDR for the elastic $s-u$-symmetric scattering amplitude of equal mass scalar particles

$$\frac{T(s,t)}{(s-2m^2+t/2)^2} = \frac{1}{2\pi i}\oint_{\mathcal{C}_s}\frac{ds'}{s'-s}\frac{T(s',t)}{(s'-2m^2+t/2)^2} = \frac{T(2m^2-t/2,t)}{(s-2m^2+t)^2} - \frac{\partial_s T(2m^2-t/2,t)}{(2m^2-s-t/2)}$$

$$+ \frac{1}{\pi}\int_{4m^2}^{\infty}\frac{ds'}{s'-s}\frac{T_s(s',t)}{(s'-2m^2+t/2)^2} + \frac{1}{\pi}\int_{-\infty}^{-t}\frac{ds'}{s'-s}\frac{T_s(s',t)}{(s'-2m^2+t/2)^2}. \tag{D.1}$$

where the contour and its deformation are given in Figure 6 with $M_{\text{gap}} = 4m^2$. In writing the formula above we assumed that the theory does not contain bound states, namely the poles located at $s < 4m^2$. If the bound states are present we subtract their contribution from the amplitude and proceed with the bound. Since such terms grow at most as $O(s)$, they do not affect the maximal growth rate of the amplitude. Note that the point $s = 2m^2 - \frac{t}{2}$ goes to itself under crossing transformation $s \to 4m^2 - s - t$. As a result due to the $s-u$ crossing, $\partial_s^{2n+1} T(2m^2-t/2,t) = 0$.

After some simple algebraic manipulations, (D.1) becomes

$$\frac{T(s,t) - T(2m^2-t/2,t)}{(s-2m^2+t/2)^2} = \frac{1}{\pi}\int_{4m^2}^{\infty}ds'\,\frac{T_s(s',t)}{(s'-2m^2+t/2)^2}\left(\frac{1}{s'-s}+\frac{1}{s'-u}\right). \tag{D.2}$$

This formula is manifestly invariant under crossing $s \to 4m^2 - s - t$.

Let us introduce a new variable for complex $s$, $s = 2m^2 - t/2 + (x+iy)$,[43] we have then

$$\frac{T(s(x,y),t) - T(s(0,0),t)}{(x+iy)^2} = \frac{2}{\pi}\int_{2m^2+t/2}^{\infty}\frac{dx'}{x'}\frac{T_s(s(x',0),t)}{x'^2 - (x+iy)^2} \tag{D.3}$$

From (D.3) and using that $T(s(0,0),t)$ is real inside the Mandelstam triangle $0 < s,t,u < 4m^2$, we see that

$$\text{Im}\,[T(s(x,y),t)] = \frac{2}{\pi}\int_{2m^2+t/2}^{\infty}\frac{dx'}{x'}\,T_s(s(x',0),t)\,\text{Im}\,\frac{(x+iy)^2}{x'^2 - (x+iy)^2} \tag{D.4}$$

$$= \frac{4}{\pi}\int_{2m^2+t/2}^{\infty}dx'\,\frac{T_s(s(x',0),t)\,xyx'}{[x'^2-(x+iy)^2][x'^2-(x-iy)^2]} \geq 0, \quad x,y \geq 0.$$

Thanks to the fact that the $y$ dependence only enters through the kernel, we can bound the local behavior of the imaginary part. After simple algebraic manipulations we get

$$-4 \leq y\partial_y \log \text{Im}\,[T(s(x,y),t)] - 1 = -\frac{\int_{2m^2+t/2}^{\infty} dx'\,T_s(s(x',0),t)\frac{4x'y^2(x^2+x'^2+y^2)}{[x'^2-(x+iy)^2]^2[x'^2-(x-iy)^2]^2}}{\int_{2m^2+t/2}^{\infty} dx'\,T_s(s(x',0),t)\frac{x'}{[x'^2-(x+iy)^2][x'^2-(x-iy)^2]}} \leq 0\,. \tag{D.5}$$

Therefore, we get the following bound on the local behavior of the amplitude

$$-3 \leq y\partial_y \log \text{Im}\,[T(s(x,y),t)] \leq 1\,, \quad 0 \leq t < m^2_{\text{gap}}. \tag{D.6}$$

In particular, the bound (D.6) states that the imaginary part of the amplitude cannot grow faster than linearly in the $y$-direction. Consider for example an amplitude that locally behaves as

$$T(s,t) \sim \lambda(t)s^2 \tag{D.7}$$

We get for it that $y\partial_y \log \text{Im}\,[T(s(x,y),t)] = 1$ and $\lambda \in \mathbb{R}$, namely that it saturates the bound on the local growth of the amplitude and it is purely elastic. From (D.4) we can conclude that $\lambda(t) > 0$.

---

[43] Here we wrote $s = s_0 + iy$ with $s_0 = 2m^2 - t/2 + x$.

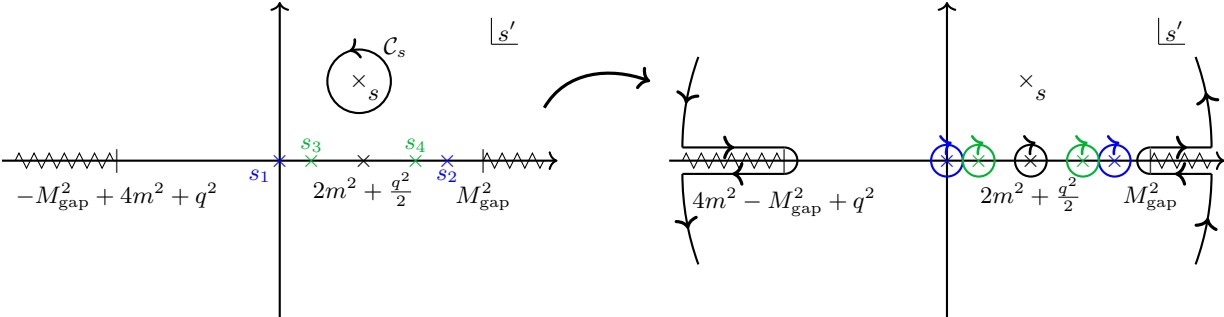

FIG. 6. The integration contour in (E.1) before and after deformation. The poles $s_1 = 0$, $s_2 = 4m^2 + q^2$ are respectively due to the $s$ and $u$-channel massless particles exchange (e.g. gravitons). The poles $s_3 = m_b^2$, $s_4 = 3m_b^2 + q^2$ are due to the exchange of massive particles in the $s$ and $u$-channel (eg. $\lambda\phi^3$ theory). These coloured poles could be absent depending on the details of the theory.

## Appendix E: Local growth bound (smeared amplitude)

In this section, we would like to reproduce the result (D.6) by considering the amplitude only in the physical regime $t \leq 0$. In this regime, we loose the positivity properties of the Legendre polynomials and we will use the smearing over transferred momenta to derive similar bounds. In particular, our analysis applies to gravitational theories in $d \geq 5$ and to theories without long-range forces (gapped or gapless) in $d = 4$. Finally, in this section, we assume the theory to be weakly coupled so that the discontinuity effectively starts at some scale $M_{\text{gap}}$ (loops of light particles are suppressed by an extra power of the small coupling).

We consider the twice-subtracted dispersion relation as in (D.1) and write

$$T_\psi(s) = \int_0^{q_0} dq \, q\psi(q) \left( s - 2m^2 - \frac{q^2}{2} \right)^2 \frac{1}{2\pi i} \oint_{\mathcal{C}_s} \frac{ds'}{s' - s} \frac{T(s', -q^2)}{\left( s' - 2m^2 - \frac{q^2}{2} \right)^2} \, , \tag{E.1}$$

where $\mathcal{C}_s$ is a small loop around $s$. In order to deform the contour, we require that the $u$-channel cut does not overlap with the origin in order to not modify the analytic structure in the $s'$ plane. This constraint implies that $q_0^2 < M_{\text{gap}}^2 - 4m^2$. Then, we can deform the contour as described in Figure 6 to obtain

$$\begin{aligned}
T_\psi(s) = &\int_0^{q_0} dq \, q\psi(q) \left[ T\left( 2m^2 + \frac{q^2}{2}, -q^2 \right) + \left( s - 2m^2 - \frac{q^2}{2} \right) \partial_s T\left( 2m^2 + \frac{q^2}{2}, -q^2 \right) \right] \\
&- \int_0^{q_0} dq \, q\psi(q) \sum_i \operatorname*{Res}_{s' = s_i} \left[ \frac{1}{s' - s} \frac{\left( s - 2m^2 - \frac{q^2}{2} \right)^2}{\left( s' - 2m^2 - \frac{q^2}{2} \right)^2} T(s', -q^2) \right] \\
&+ \int_0^{q_0} dq \, q\psi(q) \frac{1}{\pi} \int_{M_{\text{gap}}^2}^\infty ds' \, T_s(s', -q^2) \frac{\left( s - 2m^2 - \frac{q^2}{2} \right)^2}{\left( s' - 2m^2 - \frac{q^2}{2} \right)^2} \left( \frac{1}{s' - s} + \frac{1}{s' - u} \right) .
\end{aligned} \tag{E.2}$$

In the second line, the sum runs over poles of the amplitude which arise from the $s$- and $u$-channel light particles exchange, see Figure 6. We explicitly subtract these terms from the amplitude and study the local growth of the subtracted amplitude

$$\hat{T}_\psi(s) \equiv T_\psi(s) + \int_0^{q_0} dq \, q\psi(q) \sum_i \operatorname*{Res}_{s' = s_i} \left[ \frac{1}{s' - s} \frac{\left( s - 2m^2 - \frac{q^2}{2} \right)^2}{\left( s' - 2m^2 - \frac{q^2}{2} \right)^2} T(s', -q^2) \right] . \tag{E.3}$$

Not to clatter the notations below we continue to write $T_\psi(s)$, but the reader should keep in mind that the bound in the end applies to $\hat{T}_\psi(s)$. This does not affect the maximal growth rate of the amplitude because the subtraction terms are $O(s)$.

To write the last line in (E.2), we assumed that the amplitude is $s - u$ symmetric. Note that for the process $A, B \to A, B$ this is not true in general. However, it is always possible to consider the $s - u$ symmetric linear combination $T_{A,B \to A,B} + T_{A,\bar{B} \to A,\bar{B}}$ for which our results can be applied. Furthermore, due to this symmetry, $\partial_s^{2n+1} T(2m^2 - t/2, t) = 0$ and the second term in the first line can be dropped. In general, we could have considered an $s - u$ symmetric matrix of amplitudes and derive bounds on its eigenvalues. We left the detailed study of this general case for future work.

Let us now introduce the new variables for complex $s$, $s(x, y) = 2m^2 + (x + iy)$. Here we choose $s_0 = 2m^2 + x$.[44] We then obtain

$$
T_\psi(s(x,y)) = \int_0^{q_0} dq \, q\psi(q) \, T\left(2m^2 + \frac{q^2}{2}, -q^2\right)
$$
$$
+ \int_0^{q_0} dq \, q\psi(q) \frac{1}{\pi} \int_{M_{\text{gap}}^2}^\infty dx' \, T_s(s(x', 0), -q^2) \frac{\left[q^2 - 2(x + iy)\right]^2}{(2x' - q^2)[x' - (x + iy)][x' + (x + iy) - q^2]}.
$$
(E.4)

In the last line, we assumed $m \ll M_{\text{gap}}$ to simplify the lower bound of the integral. Moreover, the constraint $q_0 < M_{\text{gap}}$ implies that $0 < 2m^2 + \frac{q^2}{2} < M_{\text{gap}}^2$. Thus, the amplitude $T\left(s = 2m^2 + \frac{q^2}{2}, -q^2\right)$ in the first line of (E.4) is evaluated for real $s$ between the cuts in Figure 6. Finally, hermitian analyticity (2.2) implies that it is real. Therefore, taking the imaginary part, we see that

$$
\text{Im}\left[T_\psi(s(x,y))\right] = \int_0^{q_0} dq \, q\psi(q) \frac{1}{\pi} \int_{M_{\text{gap}}^2}^\infty dx' \, T_s(s(x', 0), -q^2) R(x, y; x', -q^2) \,,
$$
(E.5)

where for compactness we defined

$$
R(x, y; x', -q^2) = \text{Im}\left[\frac{\left[q^2 - 2(x + iy)\right]^2}{(2x' - q^2)[x' - (x + iy)][x' + (x + iy) - q^2]}\right] = \frac{(2x - q^2)(2x' - q^2)y}{[y^2 + (x - x')^2][y^2 + (x + x' - q^2)^2]} \,.
$$
(E.6)

The discontinuity of the amplitude can be decomposed in partial waves

$$
T_s(s, -q^2) = \sum_{J=0}^\infty \rho_J(s) P_J^{(d)}\left(1 - \frac{2q^2}{s - 4m^2}\right) \,,
$$
(E.7)

where $\rho_J(s) \equiv \tilde{n}_J^{(d)} \text{Im} f_J(s) \geq 0$ from unitarity. Combining everything together we obtain

$$
\text{Im}[T_\psi(s(x,y))] = \frac{1}{\pi} \int_{M_{\text{gap}}^2}^\infty dx' \sum_{J=0}^\infty \rho_J(s(x', 0)) \int_0^{q_0} dq \, q\psi(q) \left[P_J^{(d)}\left(1 - \frac{2q^2}{x' - 2m^2}\right) R(x, y; x', -q^2)\right] \,.
$$
(E.8)

Next we search for a functional for which $\text{Im} \, T_\psi(s(x,y))$ in nonnegative. In other words,

given $(x, y)$ **if** $\exists \, \psi(q)$ s.t. $\forall x' \geq M_{\text{gap}}^2$, $J = 0, 1, 2, ...$

$$
\int_0^{q_0} dq \, q\psi(q) \left[P_J^{(d)}\left(1 - \frac{2q^2}{x' - 2m^2}\right) R(x, y; x', -q^2)\right] \geq 0
$$
(E.9)

**then** $\text{Im} \, T_\psi(s(x,y)) \geq 0$ .

Thanks to the fact that the only dependence on $y$ enters through $R(x, y; x', -q^2)$, we can derive a set of constraints on the functional $\psi(q)$ such that the local bound (D.6) also holds for the smeared amplitude. The precise search

---

[44] Note that this variable is different from the one introduced in Appendix D.

algorithm is[45]

given $s(x, y)$ **if** $\exists\, \psi(q)$ s.t. $\forall x' \geq M^2,\ J = 0, 1, 2, \ldots$

$$\mathcal{C}_1(x, y; J, x') \equiv \int_0^{q_0} dq\, q\psi(q) \left[ P_J^{(d)}\left(1 - \frac{2q^2}{x' - 2m^2}\right) R(x, y; x', -q^2)\right] \geq 0$$

$$\text{and} \quad \mathcal{C}_2(x, y; J, x') \equiv \int_0^{q_0} dq\, q\psi(q) \left[ P_J^{(d)}\left(1 - \frac{2q^2}{x' - 2m^2}\right) \left(R(x, y; x', -q^2) - y\partial_y R(x, y; x', -q^2)\right)\right] \geq 0$$

$$\text{and} \quad \mathcal{C}_3(x, y; J, x') \equiv \int_0^{q_0} dq\, q\psi(q) \left[ P_J^{(d)}\left(1 - \frac{2q^2}{x' - 2m^2}\right) \left(y\partial_y R(x, y; x', -q^2) + 3R(x, y; x', -q^2)\right)\right] \geq 0$$

$$\textbf{then} \ -3 \leq y\partial_y \log \text{Im}[T_\psi(s(x, y))] \leq 1 \ .$$

$$(\text{E.10})$$

It is interesting to see what the constraints above imply at large spin and fixed impact parameter. Using that

$$\lim_{x' \to \infty} R(s, y; x', -q^2) = \frac{2(2x - q^2)y}{(x')^3} + O\left(\frac{1}{(x')^4}\right)$$

$$(\text{E.11})$$

the constraint for $x' \gg 1, J \gg 1$ becomes

$$\mathcal{C}_\infty(x, y; b) \equiv \int_0^{q_0} dq\, q\psi(q)\, (qb)^{\frac{4-d}{2}} J_{\frac{d-4}{2}}(qb) y(2x - q^2) \geq 0 \ .$$

$$(\text{E.12})$$

Other constraints from (E.10) produce the same condition after taking the limit.

For $y > 0$ and $x \gg q_0^2$ this constraint is equivalent to the positivity of the functional in the impact parameter space. By that, we mean

$$\widehat{\psi}(b) \equiv \int_0^{q_0} dq\, q\psi(q)\, (qb)^{\frac{4-d}{2}} J_{\frac{d-4}{2}}(qb)$$

$$(\text{E.13})$$

which is strictly speaking, the impact parameter transform of $\psi(q)/q^{d-4}$.

### 1. Numerical implementation

To solve the problem (E.10) of finding a functional $\psi(q)$, we linearized it by choosing the following ansatz

$$\psi_{\mathsf{a,b}}^{\{\alpha_n\}}(q) = q^{\mathsf{a}}(q_0^2 - q^2)^{\mathsf{b}} \sum_{n=0}^{N_{\max}} \alpha_n q^n$$

$$(\text{E.14})$$

We can conveniently set $M_{\text{gap}} = 1$. By our assumption, in the units of the mass gap we have $m \ll 1$ and we will effectively set the mass to zero, $m = 0$. Furthermore we choose $q_0 = M_{\text{gap}} = 1$. For the scattering of massive particles, since $q_0^2 < M_{\text{gap}}^2 - 4m^2$, if we choose the normalisation $q_0 = 1$, the gap is $M_{\text{gap}} = 1 + O(m)$.

In practice, we discretize the problem (E.10) by imposing a finite number of constraint for $J \leq J_{\max}$ and by choosing a grid for $x'$. It crucial to make a choice of grid which captures the characteristic behavior of the constraints and is sensitive to their violations. Due to the argument of the Legendre polynomials, the constraints have high frequency oscillation for small $x'$. For this reason, we chose

$$x' \in \texttt{grid x'} = \frac{1}{1 - \hat{z}} \ , \quad \hat{z} = \{0, \delta_{\hat{z}}, 2\delta_{\hat{z}}, \ldots 1 - \delta_{\hat{z}}\} \ .$$

$$(\text{E.15})$$

A similar grid was used in [15].

--------

[45] Here we wrote the algorithm for the scattering of nonidentical particles. In the case of identical particle, the constraints have to be imposed for even spins only.

Furthermore, in order to satisfy the constraints at large spin, we impose them in the impact parameter space (E.12) over a linear grid given by

$$b \in \texttt{grid b} = \{\delta_b, 2\delta_b, ...b_{\max}\} \ . \tag{E.16}$$

In total, the numerical linear problem takes the form

given $s(x,y)$ **find** $\{\alpha_n\}_{n=0,..,N_{max}}$ s.t. $\forall x' \in \texttt{grid x'}, \ J = 0, 1, ..., J_{\max}$ and $b \in \texttt{grid b}$

$$\mathcal{C}_1(x,y;J,x') \geq 0, \quad \mathcal{C}_2(x,y;J,x') \geq 0, \quad \mathcal{C}_3(x,y;J,x') \geq 0 \ \text{ and } \ \mathcal{C}_\infty(x,y;b) \geq 0 \tag{E.17}$$

**then** $-3 \leq y\partial_y \log \text{Im}[T^{\{\alpha_n\}}_{\psi_{\mathsf{a},\mathsf{b}}}(s(x,y))] \leq 1 \ .$

This problem can have many solution, in particular a functional can be found for different $N_{\max}$. In practice, we search for a *minimal* functional – with the smaller $N_{\max}$. This problem can be solved in MATHEMATICA using the function LinearProgramming. The resource consuming part is to evaluate the relevant integrals numerically.

As of now, the algorithm was described with a fixed $s(x,y)$ for simplicity. However, it is immediate to extend it to a region $s(x,y) \in A$ by choosing a grid to fill this region and impose the constraints for all point of the grid. In this work, we found it convenient to adopt a different approach. First, we find a functional for $x, y \gg 1$, then we search for its region of validity. At large $x, y \gg 1$, the problem simplifies and becomes:

**find** $\{\alpha_n\}_{n=0,..,N_{max}}$ s.t. $\forall x' \in \texttt{grid x'}, \ J = 0, 1, ..., J_{\max}$ and $b \in \texttt{grid b}$

$$\mathcal{C}_i(x \gg 1, y \gg 1; J, x') = \int_0^1 dq \ q \ \psi_{\mathsf{a},\mathsf{b}}(q) \left[ P_J^{(d)} \left( 1 - \frac{2q^2}{x'} \right) (2x' - q^2) \right] \geq 0$$

$$\textbf{and} \quad \mathcal{C}_\infty(x \gg 1, y \gg 1; b) = \widehat{\psi}(b) = \int_0^1 dq \ q \ \psi_{\mathsf{a},\mathsf{b}}(q) \ (qb)^{\frac{4-d}{2}} J_{\frac{d-4}{2}}(qb) \geq 0 \tag{E.18}$$

**then** $-3 \leq y\partial_y \log \text{Im}[T^{\{\alpha_n\}}_{\psi_{\mathsf{a},\mathsf{b}}}(s(x \gg 1, y \gg 1))] \leq 1 \ .$

Once the functional is found for $x, y \gg 1$ we determine its region of validity for the full set of constraints using (E.17).

### a. Functionals in $d = 5$

Let us now apply the algorithm defined above in five dimensions. We first find a functional for $x, y \gg 1$ using (E.18) and using numerical parameters

$$J_{\max} = 50 , \quad \delta_{\hat{z}} = \frac{1}{100} , \quad \delta_b = \frac{1}{4} , \quad b_{\max} = 40 \ . \tag{E.19}$$

Among the possible functionals, we found three simple possibility given by

$$\psi_1^{d=5}(q) = q(1-q^2)(1-q)^2 \ , \tag{E.20}$$

$$\psi_2^{d=5}(q) = q(1-q^2)(1-q)(2-q) \ , \tag{E.21}$$

$$\psi_3^{d=5}(q) = q(1-q^2)(1-q)(3-q) \ . \tag{E.22}$$

Once the functionals are found, it is possible to check analytically that they are indeed positive in the impact parameter space. These functionals may not be optimal in the sense that they may not cover the largest region in the $(x,y)$ plane. However, in the present paper, we will work with them and leave the task of finding the optimal functional for future work.

Using these functionals, we can now find the region in $(x,y)$ for which $\mathcal{C}_i(x,y;J,x') \geq 0$ and $\mathcal{C}_\infty(x,y;b) \geq 0$. In fact, as we found explicit functionals, $\mathcal{C}_\infty(x,y;b) \geq 0$ can be imposed analytically and we obtained:

$$\psi_1^{d=5}(q) : \mathcal{C}_\infty(x,y;b) \geq 0 \ \Rightarrow \ y > 0, \ x \geq \frac{7}{60} \approx 0.117 \ , \tag{E.23}$$

$$\psi_2^{d=5}(q) : \mathcal{C}_\infty(x,y;b) \geq 0 \ \Rightarrow \ y > 0, \ x \geq \frac{53}{372} \approx 0.142 \ , \tag{E.24}$$

$$\psi_3^{d=5}(q) : \mathcal{C}_\infty(x,y;b) \geq 0 \ \Rightarrow \ y > 0, \ x \geq \frac{23}{156} \approx 0.147 \ . \tag{E.25}$$

According to this constraint, it looks like $\psi_1$ can cover the largest region in the $(x,y)$-plane and we will now consider only this functional. We are left to impose $\mathcal{C}_i(x,y;J,x') \geq 0$ for $0 \leq J \leq J_{\max}$. We can do so by choosing a grid in the $(x,y)$ plane where we check numerically these constraints. However, in the case of $\psi_1^{d=5}(q)$, we found experimentally that the strongest constraints come from $J=0$. Therefore, we use the constraint $\mathcal{C}_i(x,y;J=0,x') \geq 0$ to find the region of validity of the bound. Then we numerically verify that the region holds for the higher spin constraints as well, see Figure 7.

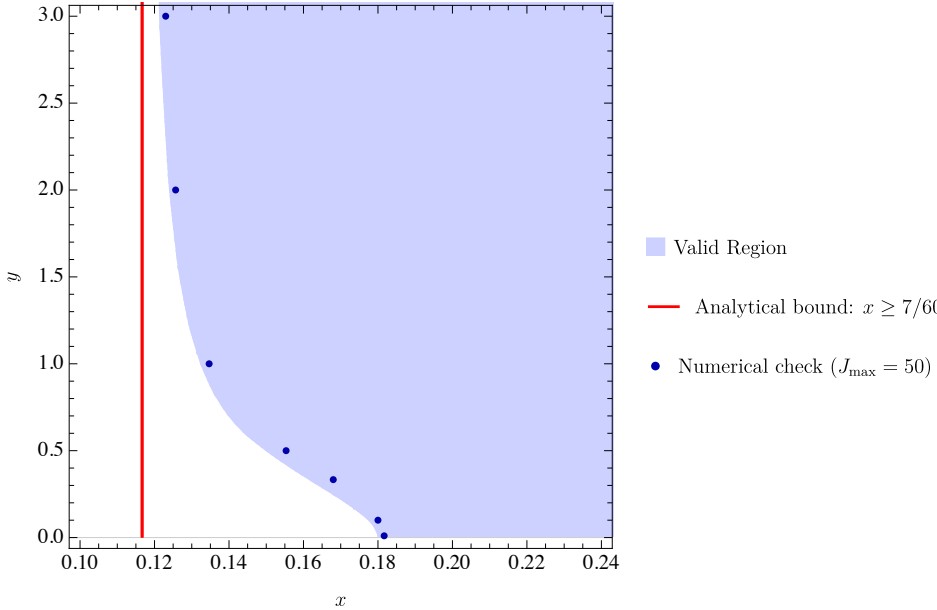

FIG. 7. Region of validity of the functional $\psi_1^{d=5}(q) = q(1-q^2)(1-q)^2$ in $d \geq 5$ in the complex $s$-plane, where $s = (x+iy)$. The region of validity was found explicitly using the $J=0$ constraint and verified on the blue dots numerically using the parameters given in (E.19). At $y \gg 1$, the valid region asymptote to the analytical bound $x \geq \frac{7}{60}$ given by the red line.

### b. Functionals in $d \geq 5$

The same algorithm can be applied in any spacetime dimension. Based on (numerical) observations, we make the following proposal

**Claim:** The functional

$$\psi_1(q) = \psi_1^{d=5}(q) = q(1-q^2)(1-q)^2 \tag{E.26}$$

obeys (E.10) in any spacetime dimensions $d \geq 5$ and the region of validity in the $s$-complex plane is the same in all dimensions (i.e. the 'Valid Region' of Figure 7).

To support this claim, first notice that the valid region in Figure 7 is determined by the $J=0$ constraint which does not depend on the dimensionality of spacetime. The positivity constraint in the impact parameter space $\mathcal{C}_\infty$ tends to be only violated for small $x$ close to $b=0$. We were not able to prove that no violation occurs at large $b$ in general dimension, but verified it in $d = 5, 6, ..., 10$, and $d = 20$. At small $b$, the $\mathcal{C}_\infty$ constraint can be explicitly computed and gives

$$A_1(d)(60x - 7)y + A_2(d)(27 - 154x)yb^2 + O(b^4) \geq 0 \ , \tag{E.27}$$

with $A_i(d) > 0$ in $d \geq 5$. Thus, we conclude that the $\mathcal{C}_\infty$ constraint is maximally violated at $b=0$ and the analytic constraint (E.23) does not depend on the dimension. Finally, for dimensions $d = 5, 6, ..., 10$ and $d = 20$, we checked numerically at the values indicated by the blue dots in Figure 7 are still in the valid region when higher spin constraints are included.

### c.   *Functionals in $d = 4$ (short-range)*

The algorithm described in (E.10) can also be used for theories in $d = 4$ without long-range forces. The novelty compared to Section D is that we only need to consider the theory in the physical region $t \leq 0$ and the mass-gap assumption can be relaxed (ie. $m_{\text{gap}} = 0$ is now allowed). However, we need to assume the theory to be weakly coupled so that we can neglect the low-energy loops. Furthermore, since we assume the absence of the $1/t$ pole (no long-range forces), we can relax the behavior close to the end points to $a > -2$, $b > -1$.

Proceeding as is Section E 1 a, we found the functional

$$\psi^{d=4}(q) = \frac{(1-q)^2}{q} \; . \tag{E.28}$$

The valid region of this functional is shown in Figure 8. Similarly as in $d \geq 5$, the strongest constraints are at $J = 0$.

While constructing the functional (E.28), we identified a family of functionals that seemingly obey (E.10) in a nontrivial region. These functionals are given by $\psi^{d=4}_n(q) = (1-q)^n/q$ with $n$ a nonnegative integer. We were not able to provide an analytic proof but checked for low $n$ that the constraints (E.18) are satisfied. For this family, the analytic bound is $x \geq \frac{1}{(2+n)(3+n)}$ which indicates a larger allowed region for large $n$. However, increasing $n$ localizes the smearing close to $t = 0$ (for which we have already derived the bound), and as such is not very suitable to constrain the scattering amplitude for $t < 0$.

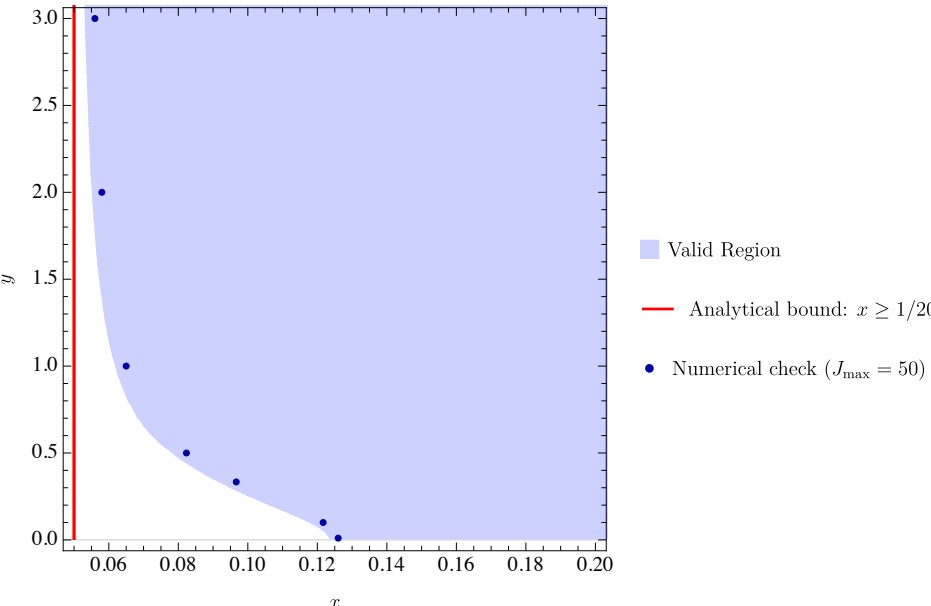

FIG. 8. Region of validity of the functional $\psi^{d=4}(q) = (1-q)^2/q$ in $d = 4$ theories without long-range forces (or $1/t$ pole). The variable $s$ is parametrized by $s = (x + iy)$. The region of validity was found explicitly using the $J = 0$ constraints and verified on the blue dots numerically using the parameters given in (E.19). At $y \gg 1$, the valid region asymptotes to the analytical bound.

### 2.   The Virasoro-Shapiro amplitude

Let us consider the four-dilaton scattering in type II superstring theory. The scattering amplitude takes the form, see e.g. [107],

$$T^{VS}(s,t) = 8\pi G_N \left( \frac{tu}{s} + \frac{su}{t} + \frac{st}{u} \right) \frac{\Gamma(1-s)\Gamma(1-t)\Gamma(1-u)}{\Gamma(1+s)\Gamma(1+t)\Gamma(1+u)}, \quad s + t + u = 0, \tag{E.29}$$

where we set $\alpha' = 4$ so that the gap in the spectrum is 1. Let us check that the amplitude satisfies the bound on the local growth that we we derived above. To apply the bound we first need to subtract the contribution of the residues

at $s, u = 0$ which gives

$$\hat{T}^{VS}(s,t) = T^{VS}(s,t) + 8\pi G_N \frac{t(2s+t)^2}{s(s+t)}. \tag{E.30}$$

Note that the subtraction term behaves as $s^0$ at large $s$ and is therefore highly sub-leading in the Regge limit. With this explicit example, we checked that indeed the local bound on scattering is satisfied for (E.30), see Figure 9. This example was built using the functional $\psi_1(q) = q(1-q^2)(1-q)^2$, valid in $d = 10$. We observe that the local growth of the Virasoro-Shapiro amplitude is consistent with our bound in its region of validity.

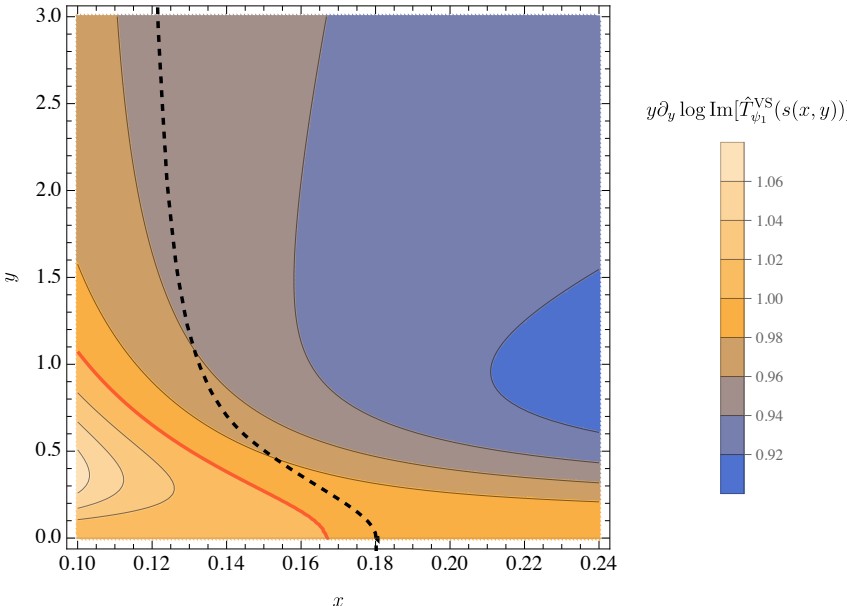

FIG. 9. Comparison of the region where the bound on the local growth of the amplitude applies (black dashed line, taken from Figure 7) using the functional $\psi_1(q) = q(1-q^2)(1-q)^2$ with the explicit example of the Virasoro-Shapiro amplitude. We observe that $y\partial_y \log \mathrm{Im} \left[ \hat{T}^{VS}_{\psi_1}(s(x,y)) \right] \leq 1$ in the region enclosed by the red line which includes the region predicted by the local growth bound. Recall that the bound is saturated if the amplitude behaves as $T(s,t) \sim \lambda(t)s^2$.

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
