# Peer review of "Gravitational Regge bounds"

_SciPost Physics_

## Round 2 · Referee Report · Francesco Giovanni Celiberto (Referee 2) · 2022-9-9

Report
They review the standard assumptions on the properties of nonperturbative amplitudes, then they imply them to gather bounds on their Regge limit.
They come out with the conclusion that scattering amplitudes admit dispersion relations that are twice-subtracted (2SDRs).
Then they show how, for a given subclass of smeared amplitudes, the gravitational amplitude for black-hole formation admits single-subtracted dispersion relations (1SDRs).
The manuscript is clearly and consistently written.
Open points and prospective developments are identified and fairly explained.
I have found particularly relevant that the 2SDR property is in agreement with AsS/CFT studies.
I agree with Authors that this represents a clear evidence of the robustness of the nonperturbative treatment.
I believe that the acceptance criteria for Scipost Physics are easily met.
The research design is appropriate and the methodology employed is adequately described.
The manuscript is also suitably formatted for publication.
I recommend the manuscript for publication in its present form.

---

## Round 2 · Referee Report · Anonymous (Referee 1) · 2022-9-9

Strengths
1) Useful review of standard assumptions 2) Very rich in results 3) Nice review on how to go to the impact parameter representation of amplitudes 4) The general logic is clearly explained
Weaknesses
1) Difficult to read 2) Many unproven technical assumptions are not sufficiently discussed 3) There is, in the referee's opinion, a poor discussion on the use of the results and ideas of the paper 4) All results are for d>4, which is not sufficiently emphasised
Report
The paper is very rich but also very technical, and presumably it is mainly directed to the experts in the field.
For what concerns the acceptance criteria of the journal, this work may "Open a new pathway in an existing or a new research direction, with clear potential for multipronged follow-up work". However, in this respect, the referee believes that the authors should make a bigger effort in explaining how their ideas could achieve this. Without such a discussion, a non-expert runs the risk of being completely cut off from the essential content of the work.
Generally speaking, it is not so easy to catch from the text what is the level of novelty of the results that are presented.
Requested changes
The referee's requests are mainly intended to make the paper more readable to the non-experts.
The authors are kindly asked to:
1) Clearly state that their results are valid for d>4, and explain what is the underlying reason in the abstract or introduction. 2) Clearly state in the introduction what are the main new results and why they are useful (also in view of point (1)). 3) Please remind the reader that J(bq) in eq. (3.16) is a Bessel function. 4) Please add a thorough introductory discussion on the "smeared bounds", either in (I.b) or in (V). This would serve as a guideline into a discussion that is quite technical.

---

## Round 2 · Referee Report · Anonymous (Referee 3) · 2022-12-29

Strengths
- Useful recapitulation of classic results on high energy gravitational scattering,
- Clearly detailed the assumptions.
- Useful lemmas on Gegenbauer polynomial estimates
- Interesting method for Regge asymptotics
- Use of various models (Born, eikonal, tidal, gravitational waves),
- Results on usual and smeared amplitudes.
Weaknesses
- Certain things could be re-written to improve readability for a reader not necessarily familiar with basics of high energy gravitational scattering (see report).
- Some results could receive more emphasis (see report).
Report
The results are about usual scattering amplitudes and smeared ones, their Regge limit and to a certain extent local growth.
Given the technicality of the subject, and of the results, a great effort as been made to make the results accessible and overall the criteria for publication in Scipost are all met.
Below, I suggest a few instances in which readability might be improved.
Requested changes
-
Given that the definition of "gravitational amplitudes" is no longer than one sentence, the authors might want to consider moving it to the very beginning, to set the stage more straightforwardly. This is just a suggestion for improving readability, and entirely up to them.
-
More of a general comment: maybe the authors would want to emphasize when comparing GW and tidal models, that G_N and alpha' (or m_tidal^2) do not enter as they only modify intermediate energies and not asymptotically large s.
-
In bullet point 4. on page 4 (Regularity), could the authors elaborate a little bit on where their expectation stems from?
-
In bullet point 5. on page 4 (Gravitational clustering), "it is the statement" -> what statement ? A forward ref to sec. III could help.
-
Above 4.16 on p.9 the author say "Our working assumption" -> isn't it also that of refs 7-10 ?
-
Is 6.3 an original result? If yes it might be good to state that fact.
-
The opening paragraph of the discussion in III.C. is a bit dry and it could help to expand it a bit by adding a few sentences.

---

## Editorial Decision

unknown